# MACHINE LEARNING FROM EXPLANATIONS

## ABSTRACT

Machine learning needs a huge amount of (labeled) data, as otherwise it might not learn the right model for different sub-populations, or even worse, they might pick up spurious correlations in the training data leading to brittle models. Also, for small training datasets, there is a huge variability in the learned models on randomly sampled training datasets, which makes the whole process less reliable. But, collection of large amount of useful representative data, and training on large datasets, are very costly. In this paper, we present a technique to train reliable classification models on small datasets, assuming we have access to some simple explanations (e.g., subset of important input features) on labeled data. We also propose a two stage training pipeline that optimizes the model's output and fine-tunes its attention in an interleaving manner, to help the model to discover the reason behind the provided explanation while learning from the data. We show that our training pipeline enables a faster convergence to better models, especially when there is a severe class imbalance or spurious features in the training data.

## 1 INTRODUCTION

Machine learning has an excellent performance in many challenging tasks (Dosovitskiy et al., 2020; Liu et al., 2021; Zhang et al., 2020), reaching or even outperforming humans (Silver et al., 2016; Li et al., 2022), in controlled experiments. However, their real-life performance is often drastically worse, especially when there are natural *spurious correlations* in training data (Arjovsky et al., 2019; Ribeiro et al., 2016), and in so many cases where the *training data set is small, heterogeneous, or unbalanced*. All this makes it hard for the existing learning algorithms to learn the right set of robust rules that can generalize well from training data. This is a serious issue in many critical applications where machine learning promises to assist humans. For example, the training data used in training human-level medical AI models often contain spurious features, and models overly rely on spurious features to classify medical images, which results in untrustworthy diagnoses in practice (Rieger et al., 2020). To make it worse, models tend to extract more spurious reasons from the smaller sets, making models more useless for minority groups (Sagawa et al., 2020). If the model does not learn the true reason, it usually finds reasons to favor the majority in the training set because it is an empirical risk minimizer. Even if the model up-weights the minority group to prevent them from being left out, which is a common practice in long-tailed classification, the model cannot classify minorities well on unseen data because the reasons extracted cannot be generalized. This discrepancy in group performance is also the base of the algorithmic (un)fairness problem in machine learning.

Getting more data seems to help to solve some obvious problems, with a significant cost (for obtaining a large amount of high-quality, accurately labeled data, and for training models on large datasets). However, it does not seem to necessarily help to learn stable and generalizable models. We observe that running the same training algorithms on the same datasets can yield models of similar accuracies that have learned drastically different decision functions and have low prediction agreements among themselves (Ross et al., 2017; D'Amour et al., 2020; Watson et al., 2022). This comes as no surprise, given the high-dimensionality of input space and parameter space, for (classification) tasks with so few labels. To fully eliminate the ambiguity in parameter space, we need datasets of huge sizes, which prohibitively increases the cost of using machine learning for most real-world problems. Collecting more data is also not always possible. For example, for (rare) disease diagnoses, there are not many new patients every year to expand the datasets. In manufacturing industries, defect detection systems for new products also do not have much data available. This shows there is a need to develop better learning algorithms in small data regimes.

In this paper, **we show that guiding machine learning with simple explanations can significantly improve its performance, reduce its sample complexity, and increase its stability**. We assume that for some training data points, in addition to their label, we also have an expert explanation for the assigned labels. The explanation can have different degrees of complexity. However, in its simplest form, we can assume that a small subset of input features is highlighted to contain the high-level reason for the assigned label. We propose an effective algorithm for machine learning with explanations, where we guide the model to identify the correct latent features most consistent with the explanations while optimizing the model's overall accuracy based on labels. Our approach significantly outperforms the baselines in convergence rate, accuracy, robustness to spurious correlations, and stability (with influences generalizability), especially for heterogeneous data. Under these settings, we show that baselines not only struggle to learn complex tasks, but also fail at properly learning the right reasons for classifying data for extremely simple tasks (e.g., detecting geometric shapes). Thus, we argue that it is *necessary* to incorporate explanations in learning algorithms if we aim at deploying trustworthy ML models that do not latch onto spurious or counter-intuitive signals.

Some prior works have explored the idea of using prior knowledge to improve machine learning (Ross et al., 2017; Rieger et al., 2020; Schramowski et al., 2020; Shao et al., 2021). Although their objective is to be "right for the *right* reasons", these methods are actually penalizing models when they learn the *wrong* reasons. As we show in our analysis, this does not necessarily result in learning the right reasons, thus it has a limited advantage to learning only from labels. This is also fundamentally different from our proposed approach. Conceptually, providing the explanation for right reasons is much easier than enumerating all possible ways that the models might make mistakes, and doing so during the data collection phase. If there are known spurious correlations, a more straightforward and more effective solution (compared to penalizing the model on learning the spurious correlations) would be to remove them and train models on clean data (Friedrich et al., 2022).

## 2 LEARNING FROM EXPLANATIONS

### 2.1 PROBLEM STATEMENT

Given a labeled dataset $D = \{(\mathbf{x}, y)_i\}$, we also have access to explanations $e(\mathbf{x}) \subset \mathbf{x}$, which are a subset of the input features, of the label for each input point $\mathbf{x}$. The explanations are informative enough that they can sufficiently explain the labels. We want to train a model to produce outputs similar to the given labels and base their decisions using reasons close to the given explanations.

### 2.2 WHY DO PREVIOUS METHODS NOT WORK WELL?

In this work, we focus on the image domain, where a wide range of model explanation methods have been studied. There are some prior works (Ross et al., 2017; Rieger et al., 2020; Schramowski et al., 2020; Shao et al., 2021) on a similar problem to ours. Explanations in their settings are bounding boxes of either the main object or the spurious features, which differ from our definition of "informative and sufficient subsets of input features". Technically, they all adopt a loss-based approach, adding an explanation misalignment loss to the label loss:

$$\mathcal{L}_{\text{joint}} = \mathcal{L}_{\text{label}} + \lambda \mathcal{L}_{\text{expl}}. \tag{1}$$

The additional loss is computed by taking the difference between the feature attributions $attr(\mathbf{x})$ computed by certain model explanation methods and the given explanation $e(\mathbf{x})$:

$$\mathcal{L}_{\text{expl}} = ||attr(\mathbf{x}) - e(\mathbf{x})||. \tag{2}$$

While it is tempting to reuse the existing algorithms for our problem, they do not lead to higher test accuracy than vanilla training in practice. The first reason is that using model explanation methods to generate models' attributions is questionable. Adebayo et al. (2018) have shown that many popular saliency map based explanations do not even pass the sanity check. Using them as a proxy of models' attention is thus unreliable. The second reason is with training. Optimization of the joint loss is often done via gradient descent. However, gradients of the two loss terms may point to different directions, creating a race condition that pulls and pushes the model into a bad local optimum. Imagine the two gradients counteract each other. The weights are then updated with negligible aggregated gradients.

This leads to models converging more slowly or not even getting updated in the worst case. Rieger et al. (2020) and Friedrich et al. (2022) have empirically verified that these methods often do not outperform the vanilla models trained with labels only, and sometimes they are strictly worse than vanilla models.

### 2.3 Tuning models' attention on latent features to learn from the explanation

To design a working algorithm that teaches the reason to models, the first step is to teach models to identify the predictive features highlighted by explanations. However, as input pixels are less semantically meaningful for image data, we need to guide the model to recognize important features in the latent space. Since explanations are sufficient, the latent representations of the given explanations should also be distinctive enough for the classifier. Hence, our core idea is to make the latent features extracted by our models more similar to the those of the explanations. We minimize the distributional difference between normalized latent features $\mathbf{x}_{\text{feat}}$ and $\mathbf{x}'_{\text{feat}}$ by optimizing the feature misalignment loss, which is the KL divergence between two normalized feature maps using softmax:

$$\mathcal{L}_{\text{feat}}(\mathbf{x}_{\text{feat}}, \mathbf{x}'_{\text{feat}}) = KL(\mathbf{x}'_{\text{feat}} \parallel \mathbf{x}_{\text{feat}}) \tag{3}$$

where $N$ is the size of the latent features. There are two important design choices in this loss function. The first one is using normalized feature maps. Without normalization, latent features can be very large or very small, potentially resulting in exploding or vanishing gradients. Secondly, we use KL divergence as the loss criterion. The main reason is that we want the distributions to be more similar, which can imply that the model's focus is primarily on the reasons region. Other loss functions may work empirically, but do not provide similar distributional intuition.

Since we also need to make models accurate, we minimize the label loss, which is the cross entropy loss between the prediction $y'$ and true label $y$:

$$\mathcal{L}_{\text{CE}}(y', y) = -y \log(y') - (1 - y) \log(1 - y'). \tag{4}$$

To avoid creating the same race condition in prior work, we propose to optimize them sequentially. Our training algorithm alternates between minimizing the label loss and the feature misalignment loss. Algorithm 1 describes our two-stage optimization.

---

**Algorithm 1** Two-stage optimization

---

**Require:** Input data $\mathbf{x}$, model $h = c \circ m \circ f$ consists of feature extractor $f$, mapping layer $m$, and fully connected layers $c$, target $y$, explanation $e(\mathbf{x})$, learning rates $\eta_1$ and $\eta_2$ for cross entropy loss and feature map loss
1: $\mathcal{L}_{\text{CE}} \leftarrow -y \log(h(\mathbf{x})) - (1 - y) \log(1 - h(\mathbf{x}))$
2: $\theta_h \leftarrow \theta_h - \eta_1 \nabla_{\theta_h} \mathcal{L}_{\text{CE}}$
3: $\mathbf{x}' \leftarrow \mathbf{x} \otimes e(\mathbf{x})$
4: $\mathbf{x}'_{\text{feat}} \leftarrow \text{softmax}(f(\mathbf{x}'))$
5: $\mathbf{x}_{\text{feat}} \leftarrow \text{softmax}(m(f(\mathbf{x})))$
6: $\mathcal{L}_{\text{feat}} \leftarrow KL(\mathbf{x}'_{\text{feat}} \parallel \mathbf{x}_{\text{feat}})$
7: $\theta_m \leftarrow \theta_m - \eta_2 \nabla_{\theta_m} \mathcal{L}_{\text{feat}}$

---

Fig 1 illustrates our two-stage training process. Given a convolutional neural network (CNN) $h$, we first decompose it into two parts: (i) the feature extractor $f$, which contains all the convolutional layers, and (ii) the classifier $c$, which contains all the fully connected (FC) layers. We insert a mapping layer $m$ in between $f$ and $c$. The mapping layer is a linear layer that maps features extracted by $f$ back to the same feature space. It will be trained to filter out irrelevant information and amplify signals from the reasons region. This addition is introduced because we do not want to update the weights of the feature extractor in the second stage. Otherwise, it would cause similar problems in prior work where two gradients counteract each other.

In the first stage, we do a normal forward pass and then backpropagate the label loss to update the entire model. In the second stage, we use the explanation $e(\mathbf{x})$ as a binary mask over the input image $\mathbf{x}$ to obtain the masked input $\mathbf{x}'$. We can then pass the masked input to the feature extractor to obtain its feature map $\mathbf{x}'_{\text{feat}} = \text{softmax}(f(\mathbf{x}'))$. We then compute the feature map $\mathbf{x}_{\text{feat}} = \text{softmax}(m(f(\mathbf{x})))$ and minimize its difference with the reasons' feature map $\mathbf{x}'_{\text{feat}}$ using Equation 3. We subsequently update the weights of the mapping layer $m$ by backpropagating the feature misalignment loss.

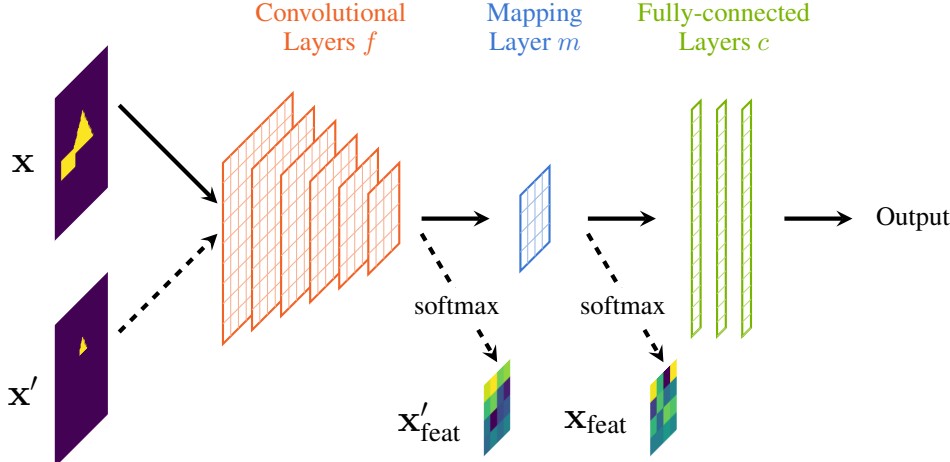

Figure 1: A demonstration of how our proposed training pipeline uses the explanation. **Solid line:** First stage of the optimization, forward pass, and backpropagation of the cross entropy loss in the same way as the conventional machine learning training pipeline. In this example, an image $\mathbf{x}$ passes through the convolutional layers, which are the feature extractor, the self mapping layer, and the fully connected (FC) layers to reach the output. The cross entropy loss (or any other loss functions) is computed, and the gradients are backpropagated to the weights. **Dotted line:** Second stage of the optimization, the new addition to the existing training pipeline that tunes the feature maps. In this diagram, the masked input $\mathbf{x}'$ based on the explanation is fed to the convolutional layers, and a feature map of $\mathbf{x}'_{\text{feat}}$ will be obtained (after applying softmax). We want the feature maps $\mathbf{x}_{\text{feat}}$ fed to the FC layers to contain as much information as $\mathbf{x}'_{\text{feat}}$. Hence we introduce a self mapping layer between the convolutional layers and the FC layers to learn a mapping function that filters out irrelevant information and extracts predictive features from the feature map. We compute the loss between the two feature maps and back-propagate the loss to the mapping layer.

## 3 EMPIRICAL ANALYSIS

We focus on binary classification with three datasets in our empirical analysis. Two of the datasets are synthetic geometric datasets that are easy for humans but challenging for machines, and one is a real dataset that is difficult even for normal humans:

- **Triangle Orientation Dataset:** If a triangle is pointing upwards, it is assigned to be Class 1. Otherwise, it is Class 0. The explanation is a rectangular area highlighting the spatial arrangement of the vertex and the bottom. (Details in Appendix A.1)

- **Fox vs Cat Dataset:** Foxes (Class 1) have triangular heads and cats (Class 0) have round heads. The explanation is a square highlighting either the vertex or the arc. (Details in Appendix A.2)

- **Bird Dataset:** We select Indigo Buntings and Blue Grosbeaks from the CUB-200-2011 dataset (Wah et al., 2011) to form our binary Bird dataset. The two species are visually identical except for the sizes of their beaks. The explanation is then a small square highlighting the beaks. (Details in Appendix A.3)

We start our empirical analysis from simple geometric datasets where a human can apply a very simple rule to perfectly classify all images. However, such simple datasets are surprisingly challenging for machine learning models if they are trained with labels only. Not only do models fail to learn the simple rule from more than enough data, but they also often pick up wrong rules that do not generalize well. We have presented the intuition of our algorithm in the previous section, so we want to empirically verify that models trained with reasons using our training pipeline are better. After observing how conventional ML training algorithms fail on the easy geometric synthetic dataset, we naturally extend our analysis to the harder Bird dataset where domain knowledge or expert advice is needed to explain the decision labels.

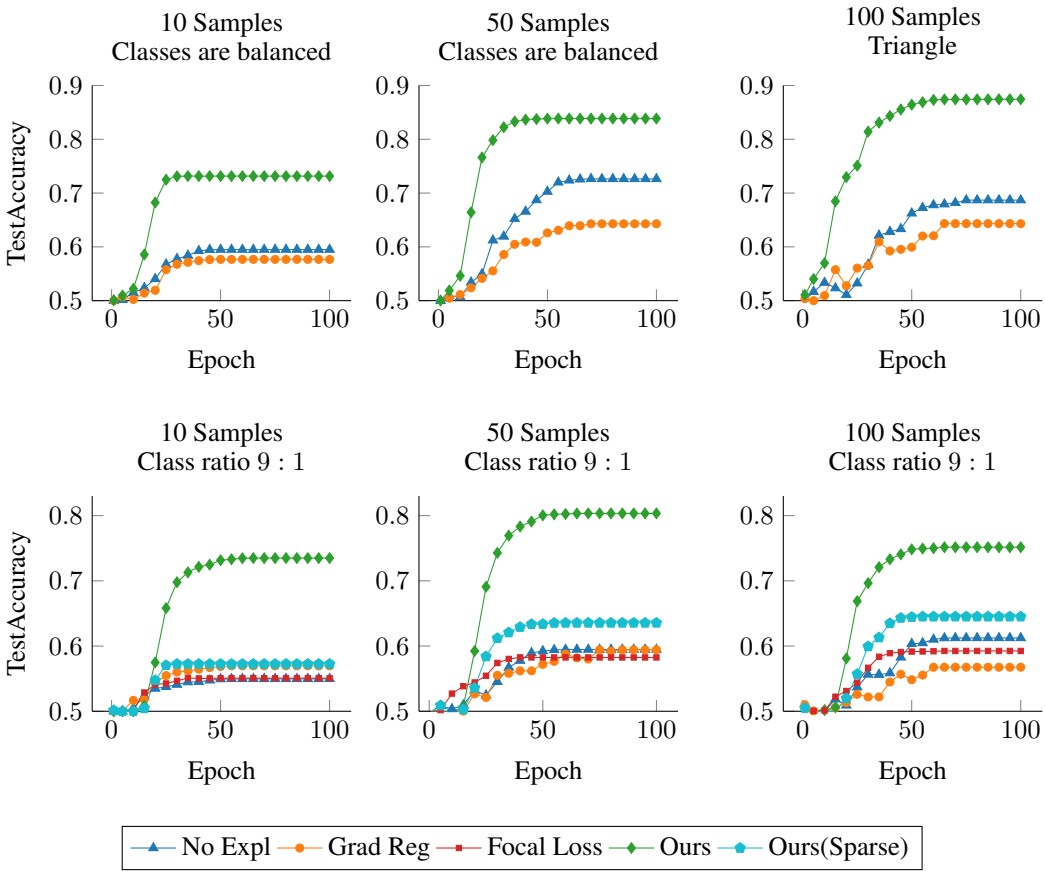

Figure 2: Test performance on the Triangle Orientation dataset. **Top row:** Training with small sample sizes with balanced classes. **Bottom row:** Training with small sample sizes when positive data only make up 10% of the training set.

We want to explore the following problems in our empirical analysis:

1. Do models benefit from learning with explanations? (Sec 3.1)

2. Do models learn the reasons suggested by the provided explanations? (Sec 3.2)

3. Does training become more consistent if we present explanations to models? (Sec 3.3)

4. Can models still learn the suggested reason when there are multiple or spurious reasons for the same decision? (Sec 3.4)

### 3.1 DO MODELS BENEFIT FROM LEARNING WITH EXPLANATIONS?

We train 30 models with early stopping for all datasets and sample sizes under each training setup. For balanced datasets, we use the vanilla training algorithm that trains with labels only. We also use the gradient regularization (Grad Reg) method proposed by Ross et al. (2017) as a baseline. For imbalanced datasets, we add two more settings. Firstly, we replace the cross entropy loss with focal loss (Lin et al., 2017), a popular loss function for imbalanced datasets, to train models with labels only. Then we consider a sparse explanation setting where we only have explanations on the minority class. We train models with the normal cross entropy loss and tune their feature maps on points with explanations. The implementation details and hyper-parameters choices can be found in Appendix B.

We observe from the top rows in Fig 2 and 3, and Fig 4 that our training pipeline can accelerate the learning process and make final models generalize better on unseen data across all datasets when classes are **balanced** in the training set. Even when there is a **severe class imbalance**, where the

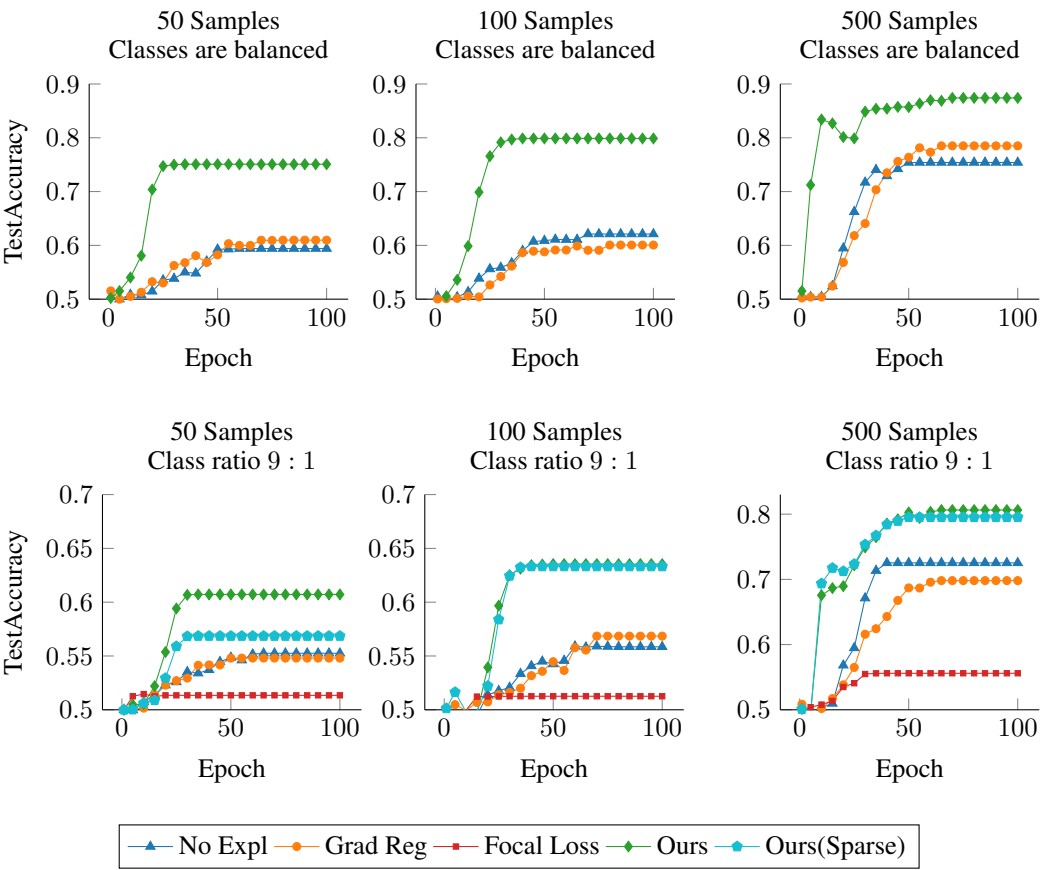

Figure 3: Test performance on the Fox vs Cat dataset. **Top row:** Training with small sample sizes with balanced classes. **Bottom row:** Training with small sample sizes when positive data only make up 10% of the training set.

class ratio in the training set is $9:1$, the bottom rows in Fig 2 and 3 show our models can reach higher test accuracy on a balanced test set. We report the standard deviations over 30 trials in Table 5 in the appendix, and we observe our models are more consistent with standard deviations of one magnitude smaller.

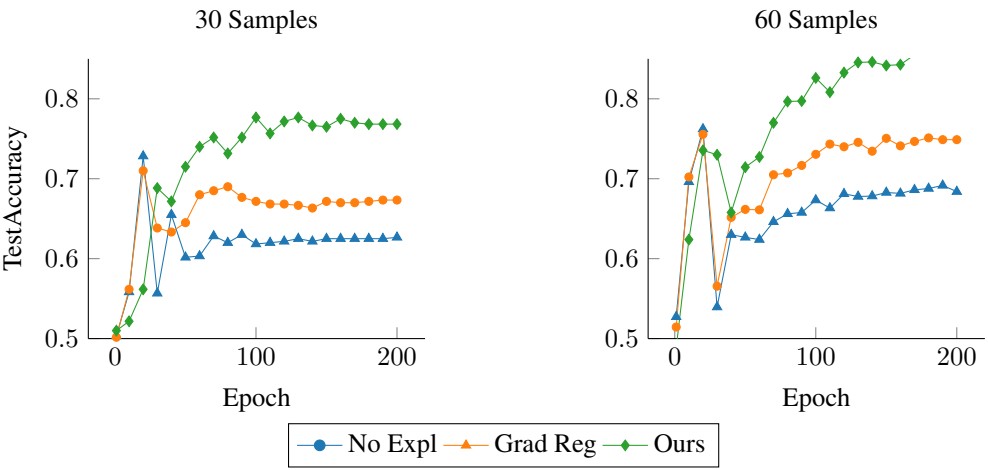

Figure 4: Test performance on the Bird dataset.

To save annotation costs, sometimes it makes sense to only query explanations on data points from the **under-represented groups**. We simulate this realistic scenario by only training with explanations when the data are from the minority class (Class 1). The cyan lines in Fig 2 and 3 demonstrate that our training strategy still outperforms others, even with sparse explanations. This is very significant given the small sample sizes in our experiments. For example, when we use a sample size of 50, the additional information only explains five images. Nevertheless, our algorithm can significantly improve the generalization performance and convergence speed. The reason for our method's usefulness in imbalanced settings without using popular re-weighting or sampling-based solutions shows that our models extract very useful features for minority groups under the guidance of explanations, so useful that the classification layers can easily distinguish the two classes in the latent space.

In addition, Fig 2, 3 and 4 reveal that the existing algorithm from Ross et al. (2017) rarely outperforms the baseline, as the orange lines are often below the blue lines. This is aligned with our claim that existing methods do not work well in practice.

## 3.2 Do models learn the reasons suggested by the provided explanations?

While we have empirically observed that presenting explanations to models helps (Section 3.1), we want to know if the superiority stems from the fact that our models learn the suggested reasons better. To analyze the alignment between the suggested reasons and the reasons models learned, previous work (Ross et al., 2017; Rieger et al., 2020; Shao et al., 2021; Schramowski et al., 2020) relies mainly on model explanation methods to visualize saliency maps and inspecting the input features with high attribution values. However, as many researchers in the community have found out, many popular model explanation methods are bad proxies of models' behavior (Adebayo et al., 2018; Shah et al., 2021). Hence, to make our analysis more convincing, we construct additional test datasets that share the same classification rules as the synthetic datasets:

- **Pentagon Orientation Dataset:** We construct this dataset by taking a vertical slice from the middle of each triangle in the test set of the Triangle Orientation dataset. The vertical slice will be a pentagon. If the triangle points upwards, the pentagon should also point upwards.

- **Triangle vs Circle Dataset:** We construct this dataset by removing the rectangles in the test set of the Fox vs Cat dataset.

Table 1: Average test accuracy of 30 models on test datasets sharing the same rules. For each row in the table, we report the test dataset we use, the training dataset, the size of the training dataset, the class ratio in the training dataset, and the test accuracy of different models obtained under different training schemes.

| Test set | Train set | Size | Class ratio | No expl | Grad | Ours | Ours(sparse) |
|---|---|---|---|---|---|---|---|
| Triangle vs Circle | Fox vs Cat | 50 | 1 : 1 | 0.587 | 0.616 | **0.772** | – |
| | | | 1 : 9 | 0.561 | 0.549 | **0.618** | 0.580 |
| | | 100 | 1 : 1 | 0.623 | 0.606 | **0.813** | – |
| | | | 1 : 9 | 0.573 | 0.574 | **0.657** | 0.642 |
| | | 500 | 1 : 1 | 0.764 | 0.795 | **0.890** | – |
| | | | 1 : 9 | 0.731 | 0.707 | **0.811** | 0.790 |
| | | 1000 | 1 : 1 | 0.846 | 0.822 | **0.916** | – |
| | | | 1 : 9 | **0.834** | 0.824 | 0.825 | 0.824 |
| Pentagon | Triangle | 10 | 1 : 1 | 0.561 | 0.565 | **0.710** | – |
| | | | 1 : 9 | 0.537 | 0.555 | **0.721** | 0.555 |
| | | 50 | 1 : 1 | 0.713 | 0.596 | **0.833** | – |
| | | | 1 : 9 | 0.573 | 0.570 | **0.799** | 0.618 |
| | | 100 | 1 : 1 | 0.673 | 0.605 | **0.855** | – |
| | | | 1 : 9 | 0.597 | 0.550 | **0.712** | 0.621 |

If models have learned the suggested reasons from the training datasets, they should be able to apply the same reasons to the additional datasets and reach high accuracy. From the results listed in Table 1, we observe that our models almost always perform the best on these datasets, suggesting our models have learned the most from the suggested reasons. Moreover, our approach outperforms the other training methods even in the sparse explanation setting. Hence, we safely claim that our models generalize better and faster because they grasp the suggested reasons better.

### 3.3 DOES TRAINING BECOME MORE CONSISTENT IF WE PRESENT EXPLANATIONS TO MODELS?

Without guidance from the reasons, models have too much freedom in choosing their decision functions. This often results in huge variances and discrepancies in models trained even with the same data. Since we have shown our models learn the given reasons better, it is important to see if they are more consistent with each other. We measure the pairwise agreement of final models by computing the percentage of identical predictions on test data. We then report the results in Table 2. Our models are always very consistent, implying they most probably use the same reasons for predictions. Models trained with labels only become more consistent when dataset size increases, supporting our hypothesis that these models can slowly infer the true reasons if the dataset is large enough.

Table 2: Average pairwise prediction agreement of 10 models when they are trained on the same dataset but with different strategies. Our models are always consistent, while models trained with labels only need large datasets to become consistent.

| Training dataset | Size | No expl | Grad Reg | Ours |
|---|---|---|---|---|
| Fox vs Cat | 50 | 0.560 | 0.600 | **0.927** |
| | 100 | 0.589 | 0.470 | **0.924** |
| | 500 | 0.782 | 0.707 | **0.944** |
| Triangle | 50 | 0.711 | 0.669 | **0.949** |
| Bird | 60 | 0.584 | 0.753 | **0.870** |

### 3.4 CAN MODELS STILL LEARN THE SUGGESTED REASON WHEN THERE ARE MULTIPLE OR SPURIOUS REASONS FOR THE SAME DECISION?

To further demonstrate the usefulness of our method, we inject **spurious** features into training data. Appendix A.4 details how we introduce the spurious features. If models can learn the right reasons to make decisions, training on spurious features should minimally affect the test performance on clean test sets. On the other hand, if models learn the spurious reasons in the training set, they will fail to generalize on the clean test set. Fig 5 shows our models are consistently the best-performing ones across all datasets and sample sizes on the clean test set, suggesting our models have learned the most out of the true reasons even when there are strong spurious signals. More results are provided in Fig 9 and Table 6 in the appendix.

## 4 RELATED WORK

Previous work on making machine learning models learn desired reasons is mostly done via regularization with a given explanation. Most papers are from natural language processing (NLP), where the attention vector is a natural source of feature attribution. Pruthi et al. (2022) trained a student model to simulate the teacher model better using attention scores. Bao et al. (2018) utilized human attention on textual data to guide machine attention. Liu & Avci (2019) treated feature attribution scores as priors to force the language models to produce similar feature attributions. Besides attention, Chefer et al. (2022) regularized relevancy scores to shift transformers' focus to the foreground. However, some

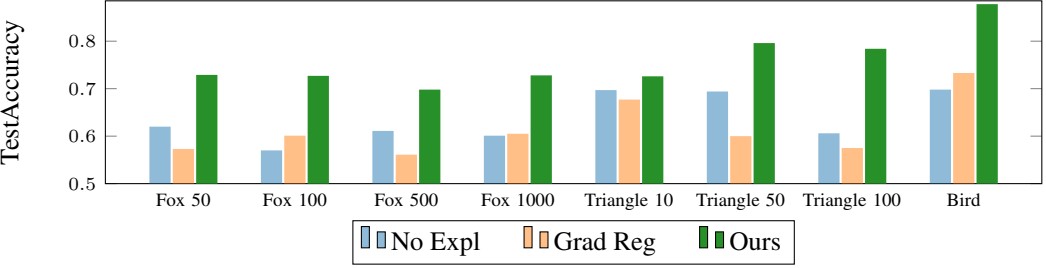

Figure 5: Test accuracy on the clean test set when models are trained on a spurious training set. Our models consistently achieve the highest accuracy, implying that they have learned more from the suggested reasons.

methods may only be useful in NLP due to the distinct characteristic of attention-based language models. Input pixels of images are also less semantically meaningful than words in language data. To our best knowledge, Ross et al. (2017) firstly explored the possibility of regularizing model explanation in training. However, their main point is that their algorithm can train diverse models focusing on different features with similar test accuracy. Du et al. (2019) focused on the alignment between model explanations and given attribution scores by optimizing the difference and enforcing sparsity, but the resulting test accuracy did not improve. Rieger et al. (2020) considered another form of explanation and focused on avoiding learning spurious correlations. Similarly, Shao et al. (2021) considered second-order derivatives as explanations and Schramowski et al. (2020) used GradCAMs. However, these methods use the knowledge of "wrong reasons" as explanations, which is fundamentally different from our approach.

## 5  DISCUSSION

We have shown that we can improve the convergence rate and generalization performance of machine learning models by inserting an intermediate linear layer and performing an additional back-propagation in each training iteration if we can query an expert for explanations. Our approach is efficient and effective, especially in a small data regime where models trained with conventional pipelines struggle to extract useful information and learn correct patterns. We also consider the usability of our approach with imbalanced data. While standard training methods require more data, particularly from the minority class, to learn the under-represented class, our method substantially improves the classification performance for the minority class when explanations are available. To make it more realistic and reduce the annotation cost, we consider the situation where only data from the minority class has explanations. We demonstrate that models trained with our algorithm can still learn faster and better, even if the additional information is only the explanation of one or five data points.

## 6  CONCLUSION

In this paper, we study a very important and useful machine learning setting: learning from explanations. We explain why previous papers fail to solve this problem and why previous loss-based methods do not bring substantial improvement. We propose a novel two-stage optimization pipeline that teaches the model to extract features according to the given explanation. We demonstrate that our algorithm significantly outperforms the current training pipeline and previous loss-based methods in both synthetic and real datasets. We also show that our models can learn the suggested reasons better and apply the reasons to applicable alternative datasets. We think our method can naturally extend to multi-class classification, and we leave it as future work. We hope that our work can inspire more researchers to adopt this training pipeline such that more real world datasets with explanations can be curated.

## 7 ETHICS STATEMENT

Our work does not have direct ethical concerns. Instead, it can benefit the under-represented groups.

## 8 REPRODUCIBILITY STATEMENT

We provide the source code our of algorithms in the submission. It comes with the default parameter values and random seeds that we use to run experiments. We also provide some details about implementation in Appendix B.

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

## A DATASETS

We use two synthetic geometric datasets and one real datasets that involves expert domain knowledge in our paper.

### A.1 TRIANGLE ORIENTATION

The first synthetic dataset is a binary classification dataset on the orientation of an triangle. A triangle is classified into class 1 if it is pointing upwards, otherwise 0. The explanation is the spatial arrangement of the vertex and the bottom line: if the vertex is above the bottom line, it is pointing upwards. The explanation is then represented as a mask highlighting the vertex and a portion of the bottom line. Each image is of grayscale with a size of 64.

## A.2 Fox vs Cat

The second synthetic dataset is a minimalistic fox vs cat dataset. A fox has a triangular head and a rectangular body, while a cat has a circular head and a rectangular body. The explanation should then focus on their heads. However, an explanation does not have to highlight the entire head region. It only needs to highlight a distinctive feature of the heads that sufficiently distinguishes a fox from a cat. Hence, the explanation for a fox is a mask highlighting the vertex of the triangular head, and the explanation for a cat is a mask highlighting the arc of the circular head (See Fig 6). Each image is of grayscale with a size of 64.

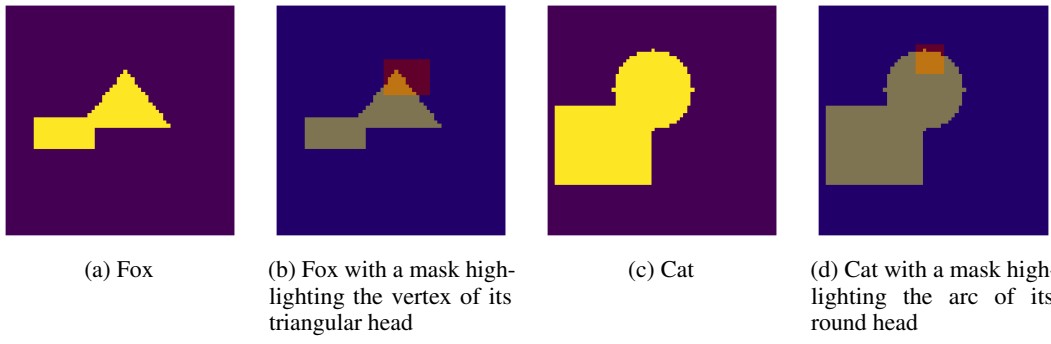

(a) Fox

(b) Fox with a mask highlighting the vertex of its triangular head

(c) Cat

(d) Cat with a mask highlighting the arc of its round head

Figure 6: Fox (Left) vs Cat (Right).

## A.3 Bird

We also extend our experiments to real datasets. We use a subset of the CUB-200-2011 (We will refer to it as the **Bird** dataset in the rest of the paper) dataset (Wah et al., 2011), a fine-grained classification dataset with photos of 200 species of birds. From the ground truth distribution of attributes for each individual class, we identify two species that are most similar to each other: the Indigo Bunting and the Blue Grosbeak (See Fig 7a and 7c). If we refer to their most likely attributes from the ground truth distribution, they are indistinguishable as they share the identical set of attributes. However, according to a bird watcher website [1], Blue Grosbeaks have bigger and heavier beaks. Hence, the explanations for our Indigo Bunting vs Blue Grosbeak classification are masks highlighting the beaks if they are visible (See Fig 7b and 7d). We use the ground truth bounding box of birds to crop all images, before resizing them to the same size for training.

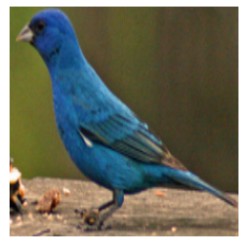 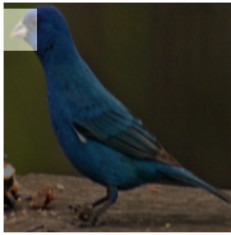 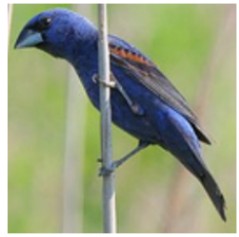 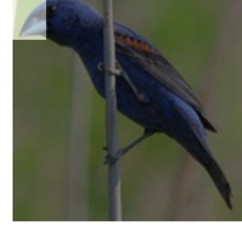

(a) An Indigo Bunting

(b) An Indigo Bunting with an explanation mask on its beak

(c) A Blue Grosbeak

(d) A Blue Grosbeak with an explanation mask on its beak

Figure 7: Indigo Buntings (Left) vs Blue Grosbeaks (Right). The difference is that Blue Grosbeaks have larger breaks. The explanations hence highlight their beaks.

## A.4 Injecting spurious features

For the Fox vs Cat dataset, we add spurious features at fixed locations. For foxes, we add a square to the top left corner; for cats, we add a square to the top right corner. For the Triangle Orientation

---

[1] https://www.sdakotabirds.com/diffids/blue_grosbeak_bunting.htm

dataset, we add a square to a random location on the leftmost column for positive data. We add the square to a random location on the rightmost column for negative data. For the CUB bird dataset, we add a green square to a random place to the negative class, and add a red square to a random place to the positive class. We make sure that these spurious features do not block the reasons masks. However, they can be arbitrarily close to the reasons.

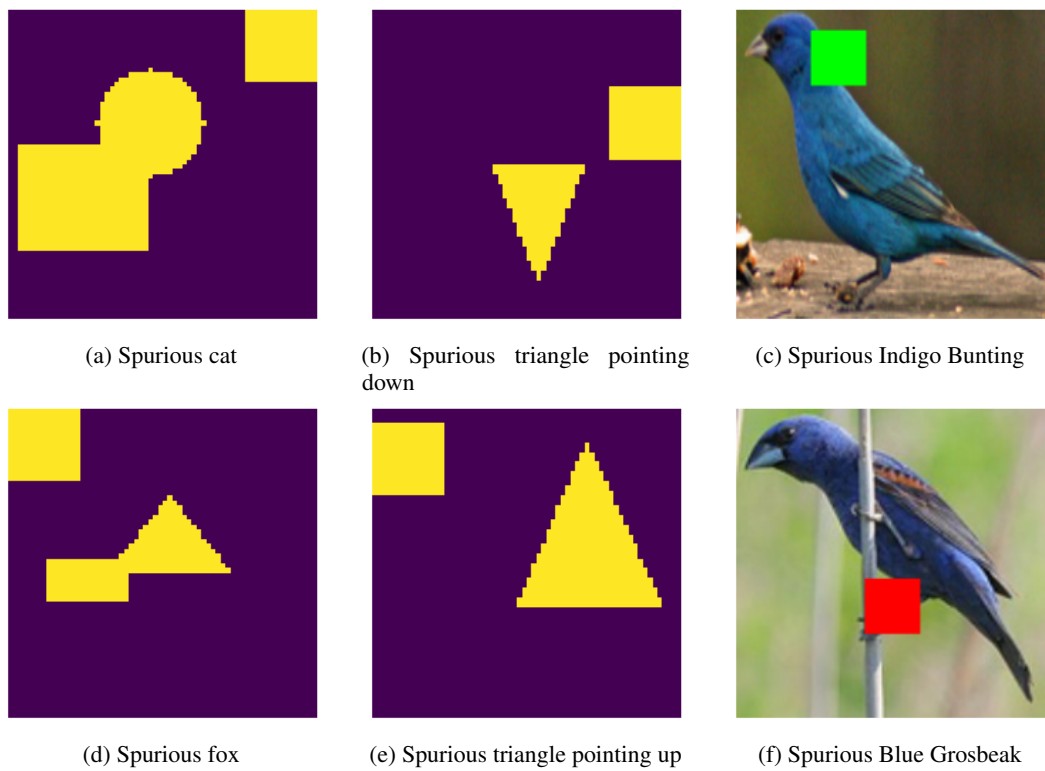

(a) Spurious cat

(b) Spurious triangle pointing down

(c) Spurious Indigo Bunting

(d) Spurious fox

(e) Spurious triangle pointing up

(f) Spurious Blue Grosbeak

Figure 8: **Top row:** images with spurious features injected from the negative class. **Bottom row:** images with spurious features injected from the positive class. **Left:** a square is added to the top right corner for cat images and the top left corner for fox images. **Mid:** a square is added to the top left corner for triangles pointing down and the bottom left corner for triangles pointing up. **Right:** a green square is added to the lower right corner for indigo buntings, and a red square is added to the lower left corner for blue grosbeaks.

## B  IMPLEMENTATION

### B.1  SYNTHETIC DATASETS

For the two synthetic datasets, we use a CNN with 3 convolutional layers and 1 FC layer. We use Adam with a learning rate of $0.01$ when we train with no explanations. When we add the gradient misalignment loss to the joint loss function, we optimize the joint loss with Adam with the same learning rate mentioned above. We set the balancing coefficient $\lambda$ of the gradient misalignment loss to be $0.001$ in equation 1, which keeps the two loss terms of similar magnitude to ensure maximum performance. This choice of hyper-parameter is mentioned by Ross et al. (2017) and also observed by us in hyper-parameter tuning. When using our method, we empirically evaluated the model with three learning rates for both $\eta_1$ and $\eta_2$: $\{0.1, 0.01, 0.001\}$. We then pick the best combination. Note that we did not tune these hyper-parameters to a great depth, so it is very likely our model can perform even better with more tuning. In summary, both optimizers are Adam with learning rates of $0.001$ when the class ratio is balanced with no spurious correlations. When there is a class imbalance or spurious correlation in data, we increase the learning rate $\eta_2$ of the feature loss optimizer to $0.01$. The

empirical evidence that $\eta_2$ needs to be increased in these two scenarios is intuitive, because models need to update their focus more to generalize well to class balanced test set.

## B.2 BIRD DATASET

For the bird dataset, we use a simple CNN with 2 convolutional layers and 1 FC layer. This is simpler than the model used in synthetic datasets because the size of training set is smaller for this dataset. To reduce the chance of overfitting, we make the model simpler. We use SGD as the optimizer for this dataset. For the baseline method without explanations, the learning rate is $0.01$. For GradReg, the balancing coefficient is computed to make the two loss terms of similar magnitudes to ensure best performance. Similar to the hyper-parameter selection procedure explained above, we pick the best combination of $\eta_1$ and $\eta_2$ from empirical results. Across all settings, $\eta_1$ is always set to be $0.001$. When the class ratio is balanced with no spurious correlations, we set $\eta_2$ to be $0.001$. On the other hand, when there is a class imbalance or spurious correlation in data, we increase the learning rate $\eta_2$ of the feature loss optimizer to $0.01$.

## C  ABLATION STUDIES

There are two novel designs in our algorithm:

1. The mapping layer
2. Two-stage optimization

We conduct ablation studies to see if any new designs bring significant improvement. Table 3 shows that the combination of the two new designs yields the best final performance and fastest convergence rate, especially on the Bird dataset. With the mapping layer, the feature difference loss is only backpropagated to the mapping layer; otherwise it will be backpropagated to all convolutional layers. As argued before, updating the convolutional layers twice with different gradients may be less effective, and the two gradients might counteract each other. On the other hand, doing joint training is conceptually inferior to our two-stage training. The only difference would be that the feature difference loss in the joint optimization scheme will be computed before the feature extractor is updated by the label loss. This is less ideal because the gradient of feature difference loss might be outdated compared to the gradient in the two-stage training scheme, so the mapping layer will be less tuned. This inefficiency results in the slower convergence rate displayed in Table 3..

## D  COST ANALYSIS

In this section, we analyze the benefit of training with explanations from the cost perspective. First of all, Table 4 shows that our method leads to lower sample complexity as the dataset size needed to reach a certain performance threshold is much smaller than that if using labels only. The table also shows that our method can achieve at least a 3x reduction in sample sizes, sometimes as much as 10x if the sample size is very small. It takes an Amazon Turk worker 1.2 cents [2] to label an image, and 3.6 cents to provide bounding boxes. The explanation used in our method is similar to bounding boxes where an annotator is only required to highlight an important region in the image. Hence the cost to obtain explanations for each image would be approximately 3.6 cents, three times the cost of labels. However, considering how much reduction in dataset sizes our methods can achieve, we believe it is more cost-efficient to query for both labels and explanations than simply collecting more data to train models if model creators start from a set of unlabeled data.

## E  ADDITIONAL RESULTS

We present some additional empirical results in this section.

---

[2]Numbers taken from `https://aws.amazon.com/sagemaker/data-labeling/pricing/`

Table 3: Model performance with standard deviation of two-stage training with mapping layer, two-stage training without mapping layer, and joint training with mapping layer. The numbers are computed at 20 epochs, 40 epochs and convergence on synthetic datasets, and at 20 epochs, 80 epochs and convergence on Bird datasets because models take longer time to converge.

| Dataset | Size | Ours | No Mapping Layer | Joint Opt |
|---|---|---|---|---|
| Fox vs Cat | 50 | $0.704 \pm 0.051$
$0.751 \pm 0.029$
$0.750 \pm 0.029$ | $0.567 \pm 0.089$
$0.686 \pm 0.076$
$0.721 \pm 0.062$ | $0.632 \pm 0.099$
$0.749 \pm 0.051$
$0.749 \pm 0.051$ |
| | 100 | $0.699 \pm 0.095$
$0.799 \pm 0.019$
$0.799 \pm 0.019$ | $0.599 \pm 0.101$
$0.691 \pm 0.094$
$0.733 \pm 0.078$ | $0.683 \pm 0.105$
$0.783 \pm 0.071$
$0.788 \pm 0.072$ |
| | 500 | $0.801 \pm 0.080$
$0.854 \pm 0.030$
$0.874 \pm 0.012$ | $0.780 \pm 0.081$
$0.801 \pm 0.067$
$0.803 \pm 0.108$ | $0.605 \pm 0.155$
$0.760 \pm 0.159$
$0.796 \pm 0.135$ |
| | 1000 | $0.738 \pm 0.158$
$0.860 \pm 0.093$
$0.903 \pm 0.017$ | $0.698 \pm 0.155$
$0.797 \pm 0.167$
$0.909 \pm 0.016$ | $0.664 \pm 0.160$
$0.756 \pm 0.167$
$0.834 \pm 0.034$ |
| Triangle | 10 | $0.682 \pm 0.110$
$0.732 \pm 0.082$
$0.732 \pm 0.082$ | $0.681 \pm 0.121$
$0.703 \pm 0.124$
$0.703 \pm 0.124$ | $0.557 \pm 0.089$
$0.627 \pm 0.112$
$0.627 \pm 0.112$ |
| | 50 | $0.766 \pm 0.100$
$0.836 \pm 0.026$
$0.839 \pm 0.028$ | $0.552 \pm 0.077$
$0.774 \pm 0.088$
$0.796 \pm 0.052$ | $0.753 \pm 0.111$
$0.827 \pm 0.060$
$0.848 \pm 0.041$ |
| | 100 | $0.730 \pm 0.115$
$0.843 \pm 0.043$
$0.874 \pm 0.018$ | $0.577 \pm 0.106$
$0.778 \pm 0.089$
$0.801 \pm 0.076$ | $0.731 \pm 0.113$
$0.843 \pm 0.051$
$0.871 \pm 0.044$ |
| Bird | 30 | $0.562 \pm 0.088$
$0.672 \pm 0.101$
$0.777 \pm 0.044$ | $0.495 \pm 0.053$
$0.530 \pm 0.091$
$0.522 \pm 0.038$ | $0.500 \pm 0.007$
$0.515 \pm 0.024$
$0.545 \pm 0.057$ |
| | 60 | $0.734 \pm 0.058$
$0.797 \pm 0.085$
$0.860 \pm 0.079$ | $0.570 \pm 0.087$
$0.692 \pm 0.069$
$0.783 \pm 0.022$ | $0.515 \pm 0.039$
$0.613 \pm 0.089$
$0.772 \pm 0.039$ |

Table 4: Test performance of models trained with no explanations and our method under different training set sizes.

| Dataset | Size | No expl | Ours |
|---|---|---|---|
| Fox vs Cat | 50 | 0.594 | 0.750 |
| | 100 | 0.621 | 0.799 |
| | 300 | 0.707 | 0.848 |
| | 500 | 0.754 | 0.874 |
| | 1000 | 0.837 | 0.903 |
| | 1500 | 0.881 | 0.920 |
| Triangle | 10 | 0.595 | 0.732 |
| | 50 | 0.726 | 0.839 |
| | 100 | 0.687 | 0.874 |
| | 200 | 0.827 | 0.900 |
| | 500 | 0.863 | 0.913 |

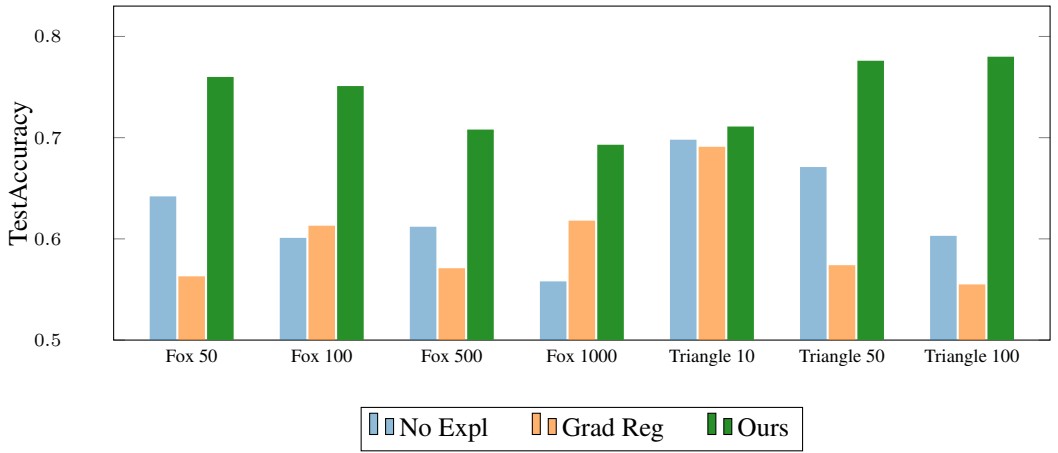

Figure 9: Test accuracy on test sets sharing the same true reasons when models are trained on spurious training set. Our models always do better, suggesting they have learned more of the true reasons.

Table 5: Black numbers: Final models performance on clean test sets when they are trained under different settings. Red numbers: Final model performance on additional datasets sharing the same reasons with the training datasets when models are trained with balanced datasets. Blue numbers: Final model performance on balanced testsets when the class ratio in the training set is $1:9$. Numbers in **bold** are best performances in each setting.

| Dataset | Size | No expl | Grad Reg | Ours |
|---|---|---|---|---|
| Fox vs Cat | 50 | $0.594 \pm 0.119$ | $0.610 \pm 0.110$ | $\mathbf{0.750 \pm 0.029}$ |
| | | $0.587 \pm 0.120$ | $0.616 \pm 0.122$ | $\mathbf{0.772 \pm 0.037}$ |
| | | $0.552 \pm 0.086$ | $0.548 \pm 0.084$ | $\mathbf{0.607 \pm 0.065}$ |
| | 100 | $0.621 \pm 0.119$ | $0.601 \pm 0.123$ | $\mathbf{0.799 \pm 0.019}$ |
| | | $0.623 \pm 0.136$ | $0.606 \pm 0.133$ | $\mathbf{0.813 \pm 0.033}$ |
| | | $0.559 \pm 0.094$ | $0.568 \pm 0.110$ | $\mathbf{0.635 \pm 0.069}$ |
| | 500 | $0.754 \pm 0.150$ | $0.785 \pm 0.127$ | $\mathbf{0.874 \pm 0.012}$ |
| | | $0.764 \pm 0.163$ | $0.795 \pm 0.134$ | $\mathbf{0.890 \pm 0.018}$ |
| | | $0.725 \pm 0.142$ | $0.698 \pm 0.145$ | $\mathbf{0.806 \pm 0.038}$ |
| | 1000 | $0.837 \pm 0.116$ | $0.815 \pm 0.070$ | $\mathbf{0.903 \pm 0.017}$ |
| | | $0.846 \pm 0.122$ | $0.822 \pm 0.083$ | $\mathbf{0.916 \pm 0.016}$ |
| | | $0.822 \pm 0.081$ | $0.819 \pm 0.034$ | $\mathbf{0.824 \pm 0.034}$ |
| Triangle | 10 | $0.595 \pm 0.118$ | $0.576 \pm 0.124$ | $\mathbf{0.732 \pm 0.082}$ |
| | | $0.562 \pm 0.092$ | $0.565 \pm 0.114$ | $\mathbf{0.710 \pm 0.104}$ |
| | | $0.550 \pm 0.093$ | $0.570 \pm 0.120$ | $\mathbf{0.735 \pm 0.101}$ |
| | 50 | $0.726 \pm 0.154$ | $0.649 \pm 0.134$ | $\mathbf{0.839 \pm 0.028}$ |
| | | $0.713 \pm 0.158$ | $0.596 \pm 0.122$ | $\mathbf{0.833 \pm 0.040}$ |
| | | $0.595 \pm 0.137$ | $0.594 \pm 0.124$ | $\mathbf{0.803 \pm 0.070}$ |
| | 100 | $0.687 \pm 0.175$ | $0.643 \pm 0.148$ | $\mathbf{0.874 \pm 0.018}$ |
| | | $0.673 \pm 0.176$ | $0.605 \pm 0.137$ | $\mathbf{0.855 \pm 0.034}$ |
| | | $0.612 \pm 0.142$ | $0.567 \pm 0.129$ | $\mathbf{0.752 \pm 0.101}$ |
| Bird | 30 | $0.627 \pm 0.128$ | $0.673 \pm 0.121$ | $\mathbf{0.777 \pm 0.044}$ |
| | 60 | $0.684 \pm 0.186$ | $0.749 \pm 0.177$ | $\mathbf{0.860 \pm 0.079}$ |

Table 6: Black numbers: Models performance on clean test sets when they are trained under different settings on datasets with spurious features. Red numbers: Model performance on additional datasets sharing the same reasons with the training datasets. Results show that models trained with our proposed setup perform the best across all three datasets and different dataset sizes, which means our models learn the most from the given reasons in the presence of strong spurious correlations.

| Spurious dataset | Size | No expl | Grad Reg | Ours |
|---|---|---|---|---|
| Fox vs Cat | 50 | 0.620 | 0.573 | **0.729** |
| | | 0.642 | 0.563 | **0.760** |
| | 100 | 0.570 | 0.601 | **0.727** |
| | | 0.601 | 0.613 | **0.751** |
| | 500 | 0.611 | 0.561 | **0.698** |
| | | 0.612 | 0.571 | **0.708** |
| | 1000 | 0.601 | 0.605 | **0.728** |
| | | 0.558 | 0.618 | **0.693** |
| Triangle | 10 | 0.697 | 0.677 | **0.726** |
| | | 0.698 | 0.691 | **0.711** |
| | 50 | 0.694 | 0.600 | **0.796** |
| | | 0.671 | 0.574 | **0.776** |
| | 100 | 0.606 | 0.575 | **0.784** |
| | | 0.603 | 0.555 | **0.780** |
| Bird | 60 | 0.698 | 0.733 | **0.878** |

