# OpenReview forum: "Machine Learning from Explanations"
_ICLR.cc/2023/Conference — Submitted to ICLR 2023_

### Official Review · Reviewer_ZDog · 2022-10-22

**Confidence:** 4
**Correctness:** 2
**Technical Novelty And Significance:** 2
**Empirical Novelty And Significance:** 1
**Recommendation:** 3

**Clarity, Quality, Novelty And Reproducibility:**

The paper is clear, although leaves more than desired for the reader to devine in terms of the experimental results. Multiple referrals to appendices, and some unclarities (this reader found himself referring to the code, e.g. to clarify what is meant by Appendix B “W set the balancing coefficient $\lambda$...", but did not find readily clear trace of it).

For quality, as mentioned above, the experimental rigor need to be improved.

If improved such that a explanations can be demonstrably used for the improvement in the practical setting, or within the realm of toy datasets and models, applicable theory can be showen, then there would be room for a novel significant contribution. Currently the novelty is limited.

Reproducibility given the code is adequately addressed by the code, and I commend the authors for providing it (could use some cleanup --- there is a lot of clutter such as mse loss that does not seem to be used etc.).

Finally, the paper would do well to engage more deeply with prior work.
The paper does not sufficiently engage with known methods of explainability and feature extraction. In particular, the notion of attention is central and pervasive both in natural language processing and in vision (e.g. vision transformers). Attention is structurally available both in the input (pixels) layer and in feature layers (i.e. for deeper than 1 layer transformers).


**Strength And Weaknesses:**

$\bf{Strengths:}$

The notion of using explanations in order to improve training is a very important one.
The authors offer a filtration layer and optimization method which allows assessing agreement between feature representations (which are not known in advance) by using characteristics which are assumed to be known as important for the labeling — hence explanations.

$\bf{Weaknesses:}$

This paper has several structural and generalizability weakness:

The datasets are extremely simple/limited. In itself that is not a problem (and can actually be a virtue!) if they were to provide theoretical insight (which is not demonstrated in the paper)  *or* empirical insight that is evaluated and carries over also in practical settings of interest.

However both are limited. And it is not demonstrated how these results can be generalized.
Can the authors show how they improve performance by providing explanations for (e.g. improving mis-classifed samples) cifar10/100 (at the least) and imagenet ?

Experimental rigor — the baseline no-explanation deteriorates with increasing number of samples (figure 2). What does that mean about the confidence in proper optimization of the experiments. Why is this happening? Is it overfitting? What happens with more samples where we know that performance of vanilla no-explanation should scale and improve.

Conceptual applicability — evaluated only in the small data regime. Is performance quickly surpassed in practice by no-explanation training with sufficient data? Then the practical utility is unclear.
Table 1 is showing that this may be the case?

Further, providing explanations in practice may be very costly. How much is an explanation worth — e.g. in terms of equivalent amount of unexplained data resulting in the same performance? How would this scale ?

It would be very exciting to show that few explanations can be used to significantly improve performance of large models. But this paper offers no insight into this practical domain of improving actual performance or showing why, in principle, it can be improved in large scale settings.


**Summary Of The Paper:**

The authors address the important topic explanations value for improving deep learning performance.
Specifically, the authors examine the problem of visual binary classification, where explanations in the form of attention/masking (carefully selected sunset of input pixels) are made available to the model while training.

The explanations are used by minimizing the kl divergence between the normalized explanation features and the normalized full image features passed through a linear filter/mapping layer constructed prior to the classification head of simple convolution architectures.

The relative performance of this scheme is evaluated on two synthetic and one natural (birds derived) data sets, under three scenarios:

1. The authors demonstrate that within the range of evaluation — up to few hundred samples  in the synthetic datasets and several tens in the birds dataset — there is a generalization benefit throughout the learning curve when training with the explanations. The evaluation is conducted both in balanced and unbalanced settings with explanations given to the minority class.

2. The authors examine the generalization on a modified dataset than trained on, but which shares the explanation classification rule — as a basis of evaluating if the models learned the ‘suggested rules provided by the explanation’.

3. The authors examine the effect of adding disturbances to the image which do not overlap with the explanation.


**Summary Of The Review:**

Utilizing human insight to improve model performance and/or cost (direct or indirect by way of data curation) is a very important topic, with strong ties to explainability and fairness as well.

This paper showcases simple visual cases where performance can be improved at the small scale.

However it does not show how these results can carry over either from theoretical ground or empirical demonstration into the practical setting of real tasks and scale, nor does it address the core question of ‘how better can one perform with explanations?’ (from a cost / data / performance scaling perspectives)

Significant work is required for addressing these issues, and I will be happy to upgrade my review if they are adequately resolved.

---

> ### Author Response · Authors · 2022-11-18
> **Response to Reivewer ZDog**
>
> 1. [*the datasets are extremely simple/limited*]
>
> Response regarding experiments on large real-world datasets is in a separate [post[(https://openreview.net/forum?id=UPQualDj1oo&noteId=TsHACb8M3Ms) on top.
>
> 2. [*Experimental rigor — the baseline no-explanation deteriorates with increasing number of samples (figure 2). What does that mean about the confidence in proper optimization of the experiments. Why is this happening? Is it overfitting? What happens with more samples where we know that performance of vanilla no-explanation should scale and improve.*]
>
> That is an accurate observation which Reviewer YcFy also pointed out. He suspected it was due to high standard deviation, which is the case here. The standard deviation is reported in Table 5 in the appendix. The performance degradation you pointed out is small compared to the standard deviation. The main reason for the large standard deviation is that the dataset sizes are very small, and overfitting is more severe.
>
> 3. [*Conceptual applicability — evaluated only in the small data regime. Is performance quickly surpassed in practice by no-explanation training with sufficient data? Then the practical utility is unclear. Table 1 shows that this may be the case?*]
>
> We believe with more data, no-explanation training will eventually **catch up** with our training instead of surpassing it. There will be an upper bound for the model performance on a given data distribution for a fixed model class, and our method can approach the upper bound in a time-efficient and sample-efficient way. In Figures 2 and 3, we can see that our models can reach the performance level with 10/50 samples, similar to baseline models with 100/500 samples. This shows a 10x sample complexity reduction. We report more results in Appendix D.
> We believe our method has practical utility. Firstly, it is difficult to tell if the data available is “sufficient” before training. Our methods can ensure that the final models will be better or equal to those trained without explanations, and our models converge with fewer epochs. Moreover, our method can **address unknown spurious correlation**s in data, which no-explanation training often picks up even in large datasets. In addition, there are scenarios where collecting more data is not possible. For example, in rare disease diagnoses, collecting more data from patients is difficult because there are only a few patients. Or if a factory wants to develop a defect detection algorithm for its new product, there are not too much data available. We believe our work has great practical value for the medical and manufacturing industries. But due to the nature of these industries, data are either not very publicly available or no one has collected explanations for the data yet.
>
> 4. [*How much is an explanation worth?*]
>
> I think you are considering our approach from an economic perspective. We have added a section in Appendix (Appendix D) to address this issue. Thank you for your suggestion.
>
> 5. [*some parts of the paper are unclear*]
>
> Thank you for your feedback. We have clarified them in this revision.
>
> 6. [*cleaning up code*]
>
> Thank you for pointing it out. We have cleaned it up.
>
> 7. [* the paper would do well to engage more deeply with prior work. The paper does not sufficiently engage with known methods of explainability and feature extraction*]
>
> We mainly focused on making models "right for the right reasons" in the paper. Our related work section thus focuses on papers targeting this problem. Model explanations paper are less related to our paper because they offer no insights into how to solve our problem, and we do not use any model explanation methods as a proxy of models' reasons. We have also included five papers that regularize attention scores of transformers to improve models in the related work. Regarding feature extraction, we did not do anything different from how convolutional neural networks extract features besides the addition of a linear layer for more stable optimization and a ground truth set of important features which we want to extract information from. Could you advise what the related works in feature extraction are so we can add them in our next version?

---

> > ### Comment · Reviewer_ZDog · 2022-11-23
> > **utility unadressed**
> >
> > central to the review and left unaddressed are the insight/utility of the method proposed.
> >
> > As stated core issues remain:
> > scaling --- to be clear, not economic, but rather data scaling: as the number of samples is increased, it is not *demonstrated* how quickly the gap in the presence of explanations vs vanilla (no explanation) is closed (or if indeed the method proposed is not surpassed as the authors expect). If SOTA, or real downstream tasks can be accelerated this can be interesting but is not demonstrated in a scenario of practical utility.
> >
> > This point of tasks of practical utility is put forth as an attempt to asses the generalizability of the method proposed.
> > If it is very hard to produce explanations in practice then the claimed utility is limited. The proposed cases where limited data with explanation may exist, would do well to be demonstrated by the authors.
> >
> > It is also not clear in principle, to what extent is the method advantageous relative to practical methods contending with scarce 'downstream' data, such as where strong representations (e.g. in using foundational self-supervised, abundant data models) are used to 'accelerate' downstream data efficiency in the Transfer Learning scenario.
> >
> > As it stands, this reviewer has not been convinced by the rebuttal as these core issues have not been addressed.

---

### Official Review · Reviewer_C28x · 2022-10-22

**Confidence:** 4
**Correctness:** 3
**Technical Novelty And Significance:** 2
**Empirical Novelty And Significance:** 3
**Recommendation:** 5

**Clarity, Quality, Novelty And Reproducibility:**

The writing is clear and the method is novel, at least to my knowledge. My main concerns are with the method design, significance (due to the assumption of known ground truth important features), and evaluation with respect to strong baselines.

**Strength And Weaknesses:**

--- Strengths ---

- Improving a model's training when ground truth explanations are available is a good idea. In cases where we don't have many training examples but we know which parts of the input is important, it would be ideal if we could improve the sample complexity for learning accurate models.
- The proposed approach seems straightforward to implement. It simply involves augmenting the standard cross-entropy loss (for encouraging accurate predictions) with a secondary loss that captures the internal representation's similar with all inputs and with the subset of relevant inputs.

--- Weaknesses ---

High-level issues
- The method only works when we have access to ground truth set of important features - an uncommon situation in practice.
- The evaluation focuses primarily on toy datasets.
- The method has several strange and seemingly unjustified design choices (see below).

There are a couple choices in the method's design that don't make sense to me, it would be helpful if the authors could explain these:
- Why is $\mathcal{L}_{\text{feat}}$ a KL divergence penalty based on normalized representations, rather than something simpler like a squared error loss? How would we normalize if the activation functions don't produce non-negative results, e.g., if we used ELU or GELU activations? A squared error loss seems more universally applicable, did the authors test this?
- The argument that the classifier and mapping layer can't be jointly optimized doesn't make much sense - people train networks with competing objectives all the time and the networks don't fail to train. Did the authors conduct ablation experiments to test whether performing joint rather than sequential training leads to meaningfully different results?
- Figure 1 seems to indicate that the classifier $h$ uses the mapping layer $m$, but this is somewhat unclear in Algorithm 1 - can the authors clarify? And it's strange that we compare the representation from $f$ with the representation from $m$, can the authors explain why we need $m$? Aside from a theoretical justification, did the authors conduct ablation experiments to test whether removing $m$ altogether leads to meaningfully different results?
- I follow how the method works, and I agree that it constitutes "learning from explanations," but I don't see what it has to do with "being right for the right reasons." The model is basically encouraged to have the same internal representation with or without the unimportant features. But the reasons for the model's prediction are not known (there's no explanation of the model's prediction) and are not explicitly encouraged to resemble the correct reasons. Wouldn't it make more sense to describe the approach slightly differently, perhaps as encouraging the model to automatically identify and disregard irrelevant input information? What are the authors' thoughts on this?

About the experimental comparisons:
- Among the baselines, only the GradReg one is in the same category as the proposed method (focal loss has nothing to do with leveraging knowledge about important features). Why did the authors not include the methods from Rieger et al (2020), Schramowski et al (2020) or Shao et al (2021)? There are two others that also could have been included mentioned below (Erion et al, 2021 and Chefer et al, 2022).
- The results from different methods presumably converge when given large enough training datasets. Could the authors provide results showing how large the training sets must be, either in the main text or supplement? E.g., there could be a plot showing the peak accuracy (or accuracy after 100 epochs) given different training dataset sizes.

About prior work:
- The claim that "most popular model explanations do not even pass the sanity check" in Adebayo et al (2018) seems like a bit of an exaggeration. The methods considered in that paper are a subset of gradient-based methods, but what about other popular methods like RISE, LIME, or SHAP? It's safe to say that there are many explanation methods that don't work as well as desired, but this claim should be dialed back.
- The authors present a hypothetical argument to explain the issue of training with a joint loss function accounting for prediction and explanation accuracy: "However, gradients of the two loss terms may point to different directions, creating a race condition that pulls and pushes the model into a bad local optimum. Imagine the two gradients counteract each other. The weights are then updated with negligible aggregated gradients." Many ML/DL methods involve training with multiple objectives and work just fine, why would it be especially problematic here? It seems correct that previous methods in this area don't work that well, but is there any evidence that this is the reason why? And even if it were, couldn't it be mitigated by pre-training with the prediction accuracy loss only and then turning on the explanation loss? This relates to one of my requests above for an ablation experiment.
- Referring to prior works that penalize explanations, the authors write: "Explanations in their settings are bounding boxes of either the main object or the spurious features, which differ from our definition of “informative and sufficient subsets of input features”." Actually, bounding boxes around the main object sound very similar to “informative and sufficient subsets of input features.” What's the significant difference? I suppose a segmentation mask would be more precise than a bounding box, but that's not a huge difference. Also, note that the authors actually use a bounding box around the beak in their one non-toy dataset.
- Referring to the same set of prior works, the authors write: "Although their objective is to be “right for the right reasons”, these methods are actually penalizing models when they learn the wrong reasons." What's the significant difference? Would you also argue that cross-entropy loss doesn't encourage classifiers to make the right predictions, they just penalize them for making the wrong predictions? This is not a well thought out criticism by the authors, but I suppose my bigger point is: I agree that there are differences between this work and prior work, but this is not helpful in clarifying what that difference is. The difference is that prior methods (at least those I'm aware of) penalize the explanation of the model being trained, whereas this method penalizes the internal representation in a manner that doesn't require generating any form of explanation.
- Two related works belonging to the Ross et al. 2017 category are "Improving performance of deep learning models with axiomatic attribution priors and expected gradients" by Erion et al (2021) and "Optimizing relevance maps of vision transformers improves robustness" by Chefer et al (2022).

**Summary Of The Paper:**

The paper proposes an approach to augment a ML model's training process when ground truth explanations (in this case, subsets of relevant features) are available. The approach is based on encouraging a model's learned representation to be similar with the full input and with the subset of relevant features.

**Summary Of The Review:**

The paper develops a method for training models when ground truth explanations are available. My recommendation is marginally below acceptance, but I could be swayed if the authors include the requested ablation experiments and compare with a stronger set of baselines.

---

> ### Author Response · Authors · 2022-11-18
> **Response to Reviewer C28x (1/3)**
>
> 1. [*The method only works when we have access to ground truth set of important features - an uncommon situation in practice.*]
>
> It is indeed an uncommon training setting, and we argue models trained with existing training methods (with labels only) suffer from many trust/robustness issues (described in the first paragraph of the introduction) because of the lack of “access to ground truth set of important features”, or explanations. It is also our proposed setting to address these common problems of ML models. We are proposing this new training paradigm to let people know a simple and effective training method exists if some explanations are available/curated in the training phase. Or if collecting more data is not possible, they can apply our method by querying experts for explanations.
>
> 2. [*The evaluation focuses primarily on toy datasets.*]
>
> Response regarding experiments on large real-world datasets are in a separate [post](https://openreview.net/forum?id=UPQualDj1oo&noteId=TsHACb8M3Ms) on top.
>
> 3. [*How would we normalize if the activation functions don’t produce non-negative results*]
>
> We did not apply activation functions to the feature maps. Instead, we use softmax, an exponential normalization function, to the raw feature maps to turn them into probabilistic distributions. We did not make it clear in our previous version, so we explicitly mention it in the figure, caption, pseudo-code, and main text in this revision. Thank you for pointing it out.
>
> 4. [*A squared error loss seems more universally applicable, did the authors test this?*]
>
>  I want to emphasize that our goal is to make the induced probabilistic feature distributions to be similar. Hence KL divergence is a more intuitive choice. We want the distributions to be similar instead of making the two feature maps point-wise identical. If the two distributions are similar, both distributions have high densities on certain support, which implies the feature maps emphasize the same features. Since we are computing the differences between two distributions, KL divergence is the natural choice of our loss criterion. Squared loss or other loss functions may also work empirically, but it lacks probabilistic motivation and semantics. We have updated our writing to explain our choice of the loss function immediately after equation 3.
>
> 5. [*Did the authors conduct ablation experiments to test whether performing joint rather than sequential training leads to meaningfully different results?*]
>
> I think what your mean by “joint” optimization by combing the backpropagation of equations 3 and 4 in one step. From the practical perspective, it is problematic because the joint backpropagation would update the weights of the convolutional layers with gradients of both losses, while we do not want to update them with the gradient of the feature difference loss. The disadvantage of using a joint loss in gradient descent is that you cannot freeze a specific layer with respect to half of the joint loss. The simple solution would be to back-propagate the two gradients separately to update the correct sets of weights, which is a two-step optimization in essence. We argue this optimization is conceptually less ideal than ours. The reason is the update of the mapping layer. In our two-stage optimization, the feature difference is taken after the feature extractor is updated with the gradient of the cross entropy loss. The mapping layer is then updated to tune the focus of the latest feature extractor. However, the feature difference loss is computed before the feature extractor is updated in the joint optimization scheme. As a result, the tuning effect on the feature extractor’s focus will be offset a bit due to the change in the feature extractor itself. Hence, the joint loss scheme is conceptually inferior to our two-stage optimization and problematic for implementation.
>
> 6. [*Figure 1 seems to indicate that the classifier uses the mapping layer, but this is somewhat unclear in Algorithm 1 - can the authors clarify?*]
>
> We have updated the figure and algorithm to clarify the role of $m$. In short, the classifier $h = c \circ m \circ f$. In the first optimization stage, the cross-entropy loss is backpropagated to the entire model. In the second stage, only $m$ is updated.
>
> 7. [*did the authors conduct ablation experiments to test whether removing m altogether leads to meaningfully different results?*]
>
> Yes, we observe that removing the mapping layer degrades the model performance, increases the standard deviation across different trials, and slows down the convergence rate. We have added these results in Table 3 in the appendix. Including $m$ is better because
> - training is more stable as only a linear layer is tuned in the second stage
> - the gradients for the conv layers wrt label loss and feature loss will not counteract each other so that convergence can be faster

---

> > ### Author Response · Authors · 2022-11-18
> > **Response to Reviewer C28x (2/3)**
> >
> > 8. [*I don’t see what it has to do with “being right for the right reasons.*]
> >
> > We do not think reasons can only be manifested by post-hoc model explanations, and we believe model explanations are not always good proxies of reasons. Intuitively, reasons used by an ML model are the features heavily relied on in their decision-making mechanisms. In our paper, we define reasons as a subset of important input features. Our algorithm explicitly encourages models to only extract information from the correct input features, thus making them base their decisions on the right set of input features and, ultimately right reasons. Your suggestion of describing our method as “encouraging the model to automatically identify and disregard irrelevant input information” is correct. But our method encourages models to disregard **all** irrelevant input features, which leaves models with only the right features.
> >
> > 9. [*Other baselines*]
> >
> > The same research group published an overview paper (Friedrich et al., 2022) where Table 2 shows Ross et al (2017) yield the same performance as Rieger et al.. (2020), Schramowski et al (2020), and Shao et al.. (2021), if not better. Hence, we think presenting Ross et al. (2017) is enough. This is not surprising because they are essentially the same method with different explanation heuristics. We have discussed why previous methods do not work well in Section 2.2. Erion et al. (2021) do not tackle the“ “right for the right reasons” problem. Instead, it aims for the smoothness and sharpness of the explanation. This is similar to Ismail et al. (2021), where a similar joint loss function is defined to ensure the sharpness of explanations. Ismail et al. (2021) noted that optimizing their joint loss would degrade model performance slightly compared to training with label loss only. Given the similarity of these two papers and the fact Erion et al. (2021) did not focus on improving model performance, we think it is not a stronger baseline in our experiments than Ross et al.. (2017). Chefer et al. (2022) propose a new way to explain visual transformers. It is unclear to us how it can be adapted to ConvNets.
> >
> > Reference:
> > - Felix Friedrich, Wolfgang Stammer, Patrick Schramowski, and Kristian Kersting. A typology to explore and guide explanatory interactive machine learning. arXiv preprint arXiv:2203.03668, 2022.
> > - Ismail, Aya Abdelsalam, Hector Corrada Bravo, and Soheil “eizi. “Improving deep learning interpretability by saliency guided training.” *Advances in Neural Information Processing Systems* 34 (2021): 26726-26739.
> >
> > 10. [*Could the authors provide results showing how large the training sets must be, either in the main text or supplement? E.g., there could be a plot showing the peak accuracy (or accuracy after 100 epochs) given different training dataset sizes.*]
> >
> > We provide a sample complexity and cost analysis in Appendix D. Overall, we observe a huge reduction in sample complexity if using our method.
> >
> > 11. [*Additional references*]
> >
> > We want to thank the reviewer for pointing out additional references. We have added Chefer et al (2022) to the related work section. However, since Erion et al. (2021) do not concern with model's the model’s performance/robustness, we think it is less relevant.
> >
> > 12. [*The claim that “most popular model explanations do not even pass the sanity check” in Adebayo et al. (2018) seems like a bit of an exaggeration.*]
> >
> > We have rewritten this claim in this revision, specifying saliency map-based explanations. On a side note, Ross et al. (2017) found out LIME is consistent with input gradients.
> >
> > 13. [*why is training with multiple objectives problematic here*]
> >
> > We are not arguing models trained with the joint loss cannot optimize both objectives. Instead, we are arguing the second loss, the explanation loss, cannot help improve model performance. This is related to our first reason outlined in 2.2, where we claim model explanations are sometimes bad proxies for the model’s attention. In fact, optimization of the joint loss indeed does two disjoint things: (i) improve model performance, (ii) make the chosen post-hoc explanations closer to the given explanation. This is supported by our experiment results, where we show optimizing the joint loss brings the same amount of improvement to model performance as training with labels only. On the other hand, Friedrich et al. (2022) show in Table 3 that jointly optimizing the explanation loss based on a chosen explanation method $A$ can reduce the wrong reasons scores computed using $A$, but not necessarily scores computed using another explanation method $B$.
> >
> > Our insights are that the explanation loss must penalize variables directly involved in the models’ decision-making mechanisms—for example, attention scores in transformers. In our case, we use latent features. If using other proxies, the loss cannot influence model performance much, so even pretraining first does not help.

---

> > > ### Author Response · Authors · 2022-11-18
> > > **Reponse to Reviewer C28x (3/3)**
> > >
> > > 14. [*This is not a well thought out criticism by the authors, but I suppose my bigger point is: I agree that there are differences between this work and prior work, but this is not helpful in clarifying what that difference is.*]
> > >
> > > We are arguing that prior methods are making models “right for not using **known** wrong reasons”, as most of them penalize gradients of wrong features. Models without using wrong reason 1 are not equivalent to models using right reasons because they can latch onto wrong reason 2. Hence, we argue there is a conceptual difference between what prior work does and “right for the right reasons”. The key difference between our and existing work is that we jumped out of the existing line of work that limits to joint optimization with an additional regularization term on model explanations. Instead, we discard the entire idea of regularizing model explanations and doing a joint optimization. We proposed a two-stage optimization process that alternatively makes the model “right” and makes the model recognize the “right reasons”. To our best knowledge, we are the first to use explanations (binary mask information) in this way to finetune models' feature extraction process.
> > >
> > > 15. [*What's the significant difference? I suppose a segmentation mask would be more precise than a bounding box, but that's not a huge difference. Also, note that the authors actually use a bounding box around the beak in their one non-toy dataset.*]
> > >
> > > Segmentation masks would be way more expensive to get if we ask human experts to label them. They also have the same problem of bounding boxes around the main object: they contain too much useless information. In our bird experiment, we use bounding boxes around the beak, which is the distinctive feature. However, if you use segmentation masks or bounding boxes around the entire birds, it will be less ideal. In fact, the baseline result (No expl) presented in our paper for the bird dataset is trained on images obtained by cropping the original image with the bounding box around the birds. It also eliminates the possibility models pick up anything else outside the bounding box. The gap between the baseline and our method shows the benefit of highlighting distinctive features instead of highlighting the entire object.

---

> > ### Comment · Reviewer_C28x · 2022-11-27
> > **Read response**
> >
> > Thanks to the authors for providing a detailed response to our feedback. I've read the response to my own concerns and unfortunately this isn't enough for me to change my score. A couple thoughts for future revisions:
> >
> > - I tend to agree with the other reviewers that you need real and more challenging datasets to demonstrate the effectiveness of this approach. You may end up having to make a dataset of your own, which could be laborious, but the current experiments with toy datasets are insufficient.
> > - About the KL divergence vs. MSE penalty: the motivation for using KL divergence seems to boil down to wanting the representations to have similar density rather than being identical. That's a fine idea, but it's not obvious that this is actually important, so you should still compare it to MSE. That would be much more convincing than vague claims about "probabilistic motivation and semantics."
> > - About other baselines: this is not the deciding factor in my score, but it would be nice if the authors were able to add more baselines. From my understanding, Erion et al. (2021) falls in the same category as other papers using different explanation methods (like Rieger et al. (2020), Schramowski et al. (2020), and Shao et al. (2021)), because you can set the penalty to be whatever you want - e.g., zero importance for irrelevant features. Adding one or more of these would make the experiments seem more fair, because several of the current comparisons have no chance of being competitive.
> > - About the writing concern numbered 14: after reading your response, I still don't agree with the statement in the paper. I agree that previous works are penalizing models for using the wrong reasons, but you are too: when a model lets irrelevant features affect the representation, it differs from the reference representation and you penalize it. I'm not saying you're doing the exact same thing, just that your statement doesn't help clarify the difference. Why not just say something straightforward like "previous works eliminate wrong reasons by penalizing their feature importance scores, but we eliminate them by ensuring that learned representations are invariant to them?" This would actually clarify what the difference is.
> > - About the writing concern numbered 15: it sounds like there's no real difference with other methods, you're just suggesting using more precise bounding boxes/segmentation masks. It's not like other papers are advocating for using imprecise bounding boxes, they just work with what's available, and they would surely benefit from having perfect bounding boxes as well. There's nothing wrong with what you're doing algorithmically, but this still seems like a meaningless statement and you might consider just removing it.

---

### Official Review · Reviewer_YcFy · 2022-10-25

**Confidence:** 4
**Correctness:** 3
**Technical Novelty And Significance:** 4
**Empirical Novelty And Significance:** 4
**Recommendation:** 5

**Clarity, Quality, Novelty And Reproducibility:**

Clarity: lacking in parts
Reproducibility: code not released


**Strength And Weaknesses:**

Strengths.
1. Well motivated problem.
2. Presentation is mostly clear.
3. Experimental evaluation is mostly through and convincing. I particularly like the imbalanced settings and evaluating if we can get away by providing explanations only for the minority class.


Weakness/questions.
1. There are mistakes in the method section that hindered understanding. Training is a two-step procedure, in the first step the parameters of h: (f, c) are updated and in the second step the parameters of mapper (m) are updated. If the outputs only depend on f, c (and not m), how does updating of m parameters make any difference? Cold be an error in the presentation here.
2. How are lambda1, lambda2 or any other parameters tuned? Especially for the imbalanced settings?
3. (minor) Line 2 and 8 of Alg 1 should have subtract rather than add for gradient updates. The mapping layer in Figure 1 is not consistent with writing. Does the mapping layer affect outputs at all?
4. To justify the need for a mapping layer, ablation experiments without the mapping layer should have been reported.
5. Authors report their method’s superiority on imbalanced datasets, but the algorithm is not designed to handle such settings, it is merely an accident if it so. In any case, some justification for why their method can handle imbalanced settings is expected.
6. Std deviation should be reported. Grad-Reg on Fox vs Cat with size 100 in Table 2 has inconsistent performance, which made me wonder if there is large variation in numbers.
7. Conceptually, I do not see why original and masked (through explanation) inputs should have the same representation. For example, when the explanation is the beak of a bird, then in order to extract the features of a beak we need to first identify that it’s a beak, which then requires spatial orientation with respect to other features of a bird. However, in the masked image, we may not have sufficient information to identify a beak as a beak and therefore may draw only poor feature representation. For this reason, I think it is repressive to constrain the masked and unmasked image representations as being alike.
8. Presence of self-sufficient explanations localized to only small regions seems like a strong assumption to me. To prove otherwise, authors should work with real-world datasets that can be explained this way. The toy experiments and bird classification experiments are not very convincing.


**Summary Of The Paper:**

ML models are known to latch to incidental correlations in the training data, which emphasises the need for controlling what they learn. This paper proposes a learning algorithm that extracts features for classification that can only be revealed from a specified region of pixels (explanation). For example, the explanation may identify the beak of a bird if we wish to learn a bird classifier that discriminated only based on beak shape/colour etc.

**Summary Of The Review:**

I agree that we should focus on a handful of “right reasons” instead of ignoring innumerable “wrong reasons”. The paper does a good job of proposing a method and showing that it is better than GradReg, but I have some conceptual and experiment related concerns that should be addressed.

---

> ### Author Response · Authors · 2022-11-18
> **Response to Reviewer YcFy (1/2)**
>
> We want to thank you for acknowledging the motivation of our paper and the novelty of our approach.
>
> 1. [*Confusion about the role of the mapping layer*]
>
> We think the source of confusion comes from the poor punctuation in the Input section of Algorithm 1. We meant $h$ contains three components, but we agree the presentation can lead to different interpretations. We have thus updated Figure 1 and Algorithm 1 to explicitly define $h=c \circ m \circ h$ to avoid confusion.
>
> 2. [*How are lambda1, lambda2 or any other parameters tuned? Especially for the imbalanced settings?*]
>
> We did not spend much time tuning the two learning rates. Instead, we tested with three values $\{0.1, 0.01, 0.001\}$ for each hyper-parameter and picked the best-performing pair. We have updated the details in Appendix B. We have also changed their notations to $\eta$ from $\lambda$ to avoid confusion with the balancing coefficient in Equation 1. In short, $\eta_1$ is $0.001$ across all settings when using our proposed method. When there is no spurious correlation or class imbalance, $\eta_2$ is $0.001$, otherwise $0.01$. You can see that $\eta_2$ is increased under class imbalance or the presence of spurious signals because the need to recognize the right reasons increases in these situations.
>
> 3. [*Line 2 and 8 of Alg 1 should have subtract rather than add for gradient updates. The mapping layer in Figure 1 is not consistent with the writing. Does the mapping layer affect outputs at all?*]
>
> Thank you for pointing it out. We have updated these two lines. The confusion about the role of the mapping layer has been addressed in point 1.
>
> 4. [*ablation experiments without the mapping layer should have been reported*]
>
> We report the results of the ablation study in Table 3 in the appendix.
>
> 5. [*justification for why their method can handle imbalanced settings is expected*]
>
> The reason that empirical risk minimization of the cross entropy loss results in poor performance on the under-represented groups is that the minority groups contribute little to the overall empirical performance, resulting in classifiers focusing on the majority groups and sacrificing the minority groups. Existing methods usually rely on under-sampling of majority groups, over-sampling of minority groups or unweighting in optimization to enforce the model not to sacrifice the minority groups to achieve higher empirical accuracy. On the other hand, our method handles imbalanced data from another perspective: we use expert explanations to guide models to extract the most information from the reasons region. This way, our models can extract more useful features for the classification layers. The fact that our method excels in imbalanced settings is strong evidence that our training method has made models aware of the right reasons, making them extracting representations so useful that the classification layers no longer have to sacrifice performance on minority groups for overall performance. In short, the experiments on imbalanced settings are included as a solid testament to our claim that our methods can teach models to learn more right reasons. We have updated the writing to re-iterate this point.
>
> 6. [*std should be reported*]
>
> We report the standard deviations in Table 5 in the appendix. You are correct that the variance in GradReg is big. Our models are more consistent as they have much smaller variances. This is also aligned with our results in 3.3 that our final models are more consistent in predictions.
>
> 7. [*Conceptually, I do not see why original and masked (through explanation) inputs should have the same representation.*]
>
> Firstly, we are not enforcing the two feature representations to be the same. Instead, we are **making the induced distribution the same**. The difference is that the former requires exact point-wise matching, while the latter only needs the shape of the density functions to be the same, implying that the relative information extracted from input features is the same. This is also related to our justification for using KL divergence as the loss criterion. Making the feature maps the same is indeed very repressive, so we did not follow this path.
> Regarding why we want to make the two feature representations have the same distribution, it is for the following reason. The masked inputs contain sufficient input features for the prediction, which is the reason, or the input region we want our models to rely on when making predictions. The feature representation of the masked inputs should only contain useful and sufficient latent features for the prediction, devoid of any redundant, useless, or spurious features. It is ideal for our models to extract information from the same input region, so we optimize our model’s feature extractor such that the feature representation it produces follow the same distribution as the ideal feature representation.

---

> > ### Author Response · Authors · 2022-11-18
> > **Response to Reviewer YcFy (2/2)**
> >
> > 8. [*Presence of self-sufficient explanations localized to only small regions seems like a strong assumption to me. To prove otherwise, authors should work with real-world datasets that can be explained this way. The toy experiments and bird classification experiments are not very convincing*]
> >
> > Response regarding experiments on large real-world datasets are in a separate [post](https://openreview.net/forum?id=UPQualDj1oo&noteId=TsHACb8M3Ms) on top.
> >
> > Our method does not restrict the reasons region to be small. The gist of our algorithm is that we force the feature map extracted to focus more on regions that are truly useful for the prediction task, so useless or spurious features will not be picked up by the model. The size of the reasons region does not affect our algorithm. In the extreme and potentially trivial case where all input features are important, the two feature maps will then be identical and the mapping layer will simply be an identity map. Our algorithm can still work but reduces to the baseline that trains with no explanations.
> >
> > In our experiments, the reasons region is small, which actually makes the optimization harder than the situation where the reasons region is larger because the amount of information available for prediction is sparser and it requires the model to ignore more features.
> >
> > 9. [*code not released*]
> >
> > We believe we submitted our code along with our paper.

---

> > > ### Comment · Reviewer_YcFy · 2022-11-29
> > > **Thanks for your response**
> > >
> > > I have now read the other reviews and the authors' response. Thanks for reporting std. dev. and correcting me about the code. I did not follow responses to some of my concerns but my major concern, as others have also pointed out, is with evaluation. I believe the work can be evaluated on ISIC dataset of skin cancer images with skin lesion binary masks, a decent such dataset should be publicly available if I am not wrong. The current datasets are too toyish otherwise also in relation to my concern/question 8. Although the paper has a bright idea to not use an explanation algorithm for supervising with saliency map, I stand unexcitedely and in doubt of some of their claims mostly due to limited evaluation.

---

### Official Review · Reviewer_pQVm · 2022-10-26

**Confidence:** 4
**Correctness:** 1
**Technical Novelty And Significance:** 1
**Empirical Novelty And Significance:** 1
**Recommendation:** 1

**Clarity, Quality, Novelty And Reproducibility:**

The paper makes a number of unsubstantiated claims with exaggerated language throughout. To give an example:
- "we argue that it is necessary to incorporate explanations in learning algorithms, if we aim at using machine learning in real-world scenarios"
- "we need datasets of astronomical sizes"

These claims are littered all over and need cites or need to be removed.

Code seems to be provided (but no README on how to interact with it), so there is partial reproducibility.

The paper was often unclear, specifically:
- specify that the explanations e(x) are binary masks much before the bottom of page 3.
- in the network (Figure 1), which layers are frozen and which layers are trained?
- equation 3 - are you computing the KL divergence between two *probability distributions* over features induced by the feature maps? In it's current form, equation 3 is dividing 2 tensors and take its log, which mathematically does not make any sense.

**Strength And Weaknesses:**

Strengths:
- The proposed setting of leveraging human explanations in order to increase performance is interesting.

Weaknesses:
- The main issue with this work is that the evaluation setup is not realistic at all. For an experimental paper like this, verifying its applicability on real-world datasets is important. Yet, 2 datasets are synthetically generated and only 1 is of real birds. This birds dataset, too, is very simple, in that the feature is very easily identifiable (the beak), and it is not clear if this method scales to more realistic distributions where the features are not as simple.
- Another huge issue is that experiments are only conducted at the extremely small-sample regime, up to 500 samples on the synthetic datasets of shapes and up to 60 examples on the bird dataset. No one is deploying machine learning trained on 60 samples. If the method was to train on all labeled data, and only incorporate some additional explanations, then that would be much more reasonable. But that is not what is happening here.

Advice:
- The idea of leveraging a few human annotations to increase performance is interesting, but the rest of the paper needs to be completely reworked. Here's what a great version of this paper would look like:
- Consider a suite of real-world datasets, such as those in the WILDS benchmark. Do not include any synthetic data experiments (they add no value) and report performance on the specific metric for each dataset. Another benefit of this is that experiments are run on non-binary tasks as well.
- Train on *all* available labeled data. The WILDS dataset contains training data splits. You should compare two main methods primarily: 1) the baseline of training on the labeled data, and 2) the new method of training on the labeled data, plus incorporating input mask explanation annotations for a few (say, 60) examples.
- Use modern backbone baselines (say, Resnet50 or DenseNet121) for the feature extraction layer - 3 conv layers is definitely too small for anything non-synthetic.
- I have to say that even given this version of the idea, I am skeptical this would work (lots of such robustness/domain invariance interventions have been proposed and have failed). But this is just my opinion, my advice, and the rest of this review is independent of this viewpoint.

**Summary Of The Paper:**

This paper proposes a method to train more accurate image classification models by leveraging human explanations in the form of input masks to increase test accuracy. While the pitched idea is promising, this work is fundamentally flawed in a number of ways and needs to be completely rethought.

**Summary Of The Review:**

The experimental results have several major flaws (as denoted above), and for this reason I believe there are significant structural changes that need to be made in order to make this a valuable contribution.

---

> ### Author Response · Authors · 2022-11-18
> **Reponse to Reviewer pQVm (1/2)**
>
> 1. [*This paper proposes a method to train more accurate image classification models by leveraging human explanations in the form of input masks to increase test accuracy.*]
>
> We think you might have mistaken the purpose of this paper. Making the model more accurate is not the main contribution or the main focus of this paper. The purpose of the proposed training pipeline is to guide ML models to **learn rules suggested by human explanations**. The alignment of the decision rules of ML models and human intuition is an equally important, if not more important, aspect of trustworthy machine learning than empirical accuracy. All experiments described in this paper serve the purpose of supporting our claim that our proposed pipeline can make models recognize the right reasons, including computing accuracy. Naturally, if a model can learn the correct rule, it should generalize better. Hence, we demonstrate that our models are “more accurate”. With the extra guidance, models should also converge faster, which is confirmed in the accuracy plot. We then conduct more experiments to assess whether our models recognize the right reasons more by
>   - Testing on datasets with the same rules (Section 3.2)
>   - Measuring final model consistency (Section 3.3)
>   - Evaluating robustness on spurious features (Section 3.4)
> All these experimental results are in line with our claim that our models learn the right reasons more.
>
> 2. [*The main issue with this work is that the evaluation setup is not realistic at all.*]
>
> We do not agree with your claim that our experiment setup is “unrealistic”. The reason that we start from synthetic datasets is that they are so easy that you would expect ML models to nail them with very few data points, which is not the case. Why are ML models failing to identify “easily identifiable features” on very simple datasets? Our answer is that ML models trained with labels alone are clueless about the decision rules and thus, require unnecessarily large datasets. Since the problem identified is the **inability to recognize the right reasons**, we propose a new training method to teach models the right reasons in training. The small sample size is not a bug, but a testament to how well (with how little data) our method can teach the right reasons. We do not need to train on all labeled data for our models to achieve a certain level of generalization performance.
>
> 3. [*Consider a suite of real-world datasets, such as those in the WILDS benchmark. Do not include any synthetic data experiments (they add no value) and report performance on the specific metric for each dataset. Another benefit of this is that experiments are run on non-binary tasks as well.*]
>
> We greatly appreciate the advice you provided. Extending our experiment to larger real datasets is definitely beneficial for the analysis. Response regarding experiments on large real datasets are at a separate [post](https://openreview.net/forum?id=UPQualDj1oo&noteId=TsHACb8M3Ms) on top. However, we do not agree that synthetic datasets are useless. Firstly, all previously published works on “making models right for the right reasons” rely on synthetic datasets such as Decoy-MNIST or C-MNIST. These two datasets have much simpler and more easily identifiable features than the datasets used in our paper. We also do not see why our approach cannot be extended to multi-class classification, because our algorithm does not have components that are specific for binary classification. We have also demonstrated that the reasons learned by our models are transferable to other datasets. Once again, if ML models fail to recognize the simple rules present in these synthetic datasets, it speaks volumes about why ML models fail to generalize. Therefore, we believe synthetic datasets are useful and the insights are valuable.
>
> 4. [*Exaggerated language throughout*]
>
> We have toned down the sentences you pointed out.
>
> 5. [*specify that the explanations e(x) are binary masks much before the bottom of page 3*]
>
> We mentioned in the introduction and problem statement that the explanations are (highlighted) subsets of input features, which are conceptually and practically the same as the binary mask.
>
> 6. [*which layers are frozen and which layers are trained?*]
>
> In the paragraphs below Algorithm 1, we described the two-stage optimization where the CE loss is backpropagated to all layers while the feature difference loss is only used to update the mapping layer. In the pseudo-code, Line 2 shows $h$, which is the entire network, is updated by the CE loss; Line 8 shows only the mapping layer $m$ is updated by the feature difference loss. We have updated Figure 1 by adding the notations used in Algorithm 1 to each layer ($f, m, c$ accordingly) to avoid confusion. We also made it clear that $h=c \circ m \circ f$ to further illustrate the forward pass.

---

> > ### Author Response · Authors · 2022-11-18
> > **Response to Reviewer pQVm (2/2)**
> >
> > 7. [*equation 3 is dividing 2 tensors and take its log, which mathematically does not make any sense*]
> >
> > Yes, we are computing the KL divergence between the normalized feature distributions induced by feature maps using a softmax operation. We are not dividing tensors, but each entry in the tensors. Equation 3 was exactly how this loss is computed in [PyTorch](https://pytorch.org/docs/stable/generated/torch.nn.KLDivLoss.html). To emphasize that we are using KL divergence, we updated equation 3 with the standard KL divergence notion.
> >
> > 8. [Reproducibility]
> >
> >  We have updated the code with detailed instructions on running and reproducing the experiments.

---

> > > ### Comment · Reviewer_pQVm · 2022-11-18
> > > **unconvinced**
> > >
> > > I remained unconvinced with the rebuttal. The main reason evaluation is limited is, as the authors state, "no suitable large real datasets are available". Then why don't the authors go out and collect one? That would be a valuable contribution to the community and to this area of research. The authors reasoning about prior works validating on MNIST is not convincing - anyone can get anything to work on MNIST. MNIST is not predictive of real world use. If the proposed method works well, the authors wouldn't have to collect many annotations anyway.

---

### Author Response · Authors · 2022-11-18
**On our experiment setup and practical value**

A common suggestion among reviewers is that we need to experiment on large real-world datasets to prove the practical value of our approach. We want to address it centrally here.

Firstly, we are proposing a new machine learning paradigm where we have explanations in addition to labels. The current training pipeline does not involve explanations at all, so all current datasets curated do not include explanations. We would love to verify the utility of our method on large real datasets, but given that no suitable large real datasets are available, we cannot conduct the evaluation. Our situation is similar to another [paper](https://openreview.net/forum?id=aM7UsuOAzB3&noteId=Uv_JkEdjJYF) whose setting is new and cannot find large real datasets for evaluation. In the end, the reviewers and the AC acknowledged that "the lack of large scale systematic evaluation might also be due to lack of datasets and the nature of the problem being studied".

We have demonstrated that incorporating human explanations into machine learning pipelines can be powerful in teaching models the right reasons to use. By making models focus more on the right reasons, models can converge faster, generalize better, and require fewer data. We hope our work can make researchers more aware of an alternative machine learning paradigm and inspire more researchers and practitioners to adopt our pipeline. In this way, more explanations can be collected, and more suitable datasets can be curated.

Secondly, prior works trying to solve the same problem also heavily rely on synthetic datasets. In fact, many synthetic datasets used are simpler than those in our paper. Friedrich et al. (2022) verified that vanilla training could lead to better performance once a simple data pre-processing step is done to remove the known wrong reasons. We believe our experiment results are a significant improvement over the prior works.

Lastly, regarding the practical value of our paper, we argue it is not always possible to have a large dataset for perusal. Sometimes, it is not even possible to collect more data. For example, for rare disease diagnoses, there are very few patients every year; collecting more positive data is impractical. In another instance, when creating defect detection systems for new products in the manufacturing industry, there is also a lack of data because the product has yet to be mass-produced, let alone finding many defective ones. Under these circumstances, where we need to train ML models with a fixed-size dataset with insufficient data, our method can be very useful.

Reference:
Felix Friedrich, Wolfgang Stammer, Patrick Schramowski, and Kristian Kersting. A typology to explore and guide explanatory interactive machine learning. arXiv preprint arXiv:2203.03668, 2022.

---

### Decision · Program_Chairs · 2023-01-20

**Decision:**

Reject

**Justification For Why Not Higher Score:**

Reviewers are not convinced on the practical utility of the proposed paradigm and method.

**Justification For Why Not Lower Score:**

N/A

**Metareview: Summary, Strengths And Weaknesses:**

The paper proposes a new learning paradigm to train reliable classification models on small datasets, assuming access to some simple explanations (e.g., subset of important input features) on labeled data. Experiments on several synthetic and real datasets showed promise of the proposed method. While all the reviewers consider the proposed setting of leveraging human explanations to increase model performance interesting, they are not convinced by the applicability of the proposed method in real world settings. Aiding learning with explanation is probably more helpful in small data regime. The authors however focused the experiments on image classification tasks which often has abundant training examples.  Application domains such as medical data where labeled data is hard to come by, might be able to better showcase the value of the proposed paradigm and method.